# THE DISPARATE IMPACTS OF SPECULATIVE DECODING

## ABSTRACT

The practice of speculative decoding, whereby inference is probabilistically supported by a smaller, cheaper, "drafter" model, has become a standard technique for systematically reducing the decoding time of large language models. This paper conducts an analysis of speculative decoding through the lens of its potential disparate speed-up rates across tasks. Crucially, the paper shows that speed-up gained from speculative decoding is not uniformly distributed across tasks, consistently diminishing for under-fit, and often underrepresented tasks. To better understand this phenomenon, we derive an analysis to quantify this observed "unfairness" and draw attention to the factors that motivate such disparate speed-ups to emerge. Further, guided by these insights, the paper proposes a mitigation strategy designed to reduce speed-up disparities and validates the approach across several model pairs, revealing up to a 76.7% improvement in our fairness metric.

## 1 INTRODUCTION

The rapid growth of large language models (LLMs) has motivated the search for more effective inference paradigms. Among these, speculative decoding (Leviathan et al., 2023) has emerged as the leading approach for accelerating text generation. This methodology offloads much of the token-generation work onto a lightweight "drafter" model, which proposes a set of candidate tokens to be generated. These candidates are then verified by a larger "verifier" model in parallel, which accepts or rejects them based on a specific acceptance criteria that can ensure invariance from the vanilla decoding process. When streaks of tokens are accepted by the verifier, an inference speed-up is achieved. The effectiveness of speculative decoding fundamentally depends on the alignment between the drafter and verifier conditional token distributions. When the drafter and target conditional distributions are well-aligned, acceptance rates are high and throughput gains are substantial. Conversely, misalignment sharply reduces acceptance and erodes speed-up.

This dependency has motivated a line of work on designing better drafters and verification schemes, e.g., distillation to improve alignment (Zhou et al., 2023) or structural changes that increase acceptance (Li et al., 2024). Yet these advances optimize *average* throughput and say little about how acceleration is distributed across tasks or user groups. This gap matters in many application contexts, with particular relevance for multilingual deployment. Multilingual use is a key driver of LLM adoption, but tokenization non-uniformities and data imbalance can conspire to make some languages systematically "harder" for both drafting and verification (Petrov et al., 2023b). For example, even when downstream models are identical, subword tokenizers can induce order-of-magnitude differences in sequence lengths across languages, with direct implications for per-request latency and cost budgets (Petrov et al., 2023a). This raises an important question: *Do certain tasks systematically experience lower speed-ups than others in the context of speculative generation, and can these speed-up disparities be modeled and corrected?* This paper investigates this *computational unfairness* phenomenon and reveals a pattern predictive of systematic slowness: *Tasks to which the drafter exhibits relatively less fitness tend to suffer from lower speed-ups*. This creates a fairness issue where the efficacy of speculative decoding at accelerating inference becomes unevenly distributed across tasks.

**Key contributions.** To address this phenomenon, this study presents several contributions. First, given next token distributions $p(x), q(x)$, for verifier and drafter models respectively, the paper establishes monotonic links between speculative decoding speed-up, $S$, and different notions of model divergence (i.e., total variation, $\mathrm{TV}(p(x), q(x))$ and cross-entropy, $\mathrm{H}(p(x), q(x))$). Second, based on these results, the paper defines an optimizable and justified notion of speculative decoding

unfairness, $\mathcal{U}$, built on the smooth divergence function $H(p(x), q(x))$. Next, the paper introduces an analysis that highlights the connections between drafter-fitness and speculative speed-up, establishing disparities in drafter fitness as a predictive source of unfairness. Finally, it showcases the consistent prevalence of speed-up unfairness in a variety of settings and introduces a justified mechanism to mitigate acceleration disparities across tasks, denoted stochastic corrective drafter finetuning (s-CDF).

The results of this work show that speculative decoding could be a potential source of *computational inequity* where some tasks or communities could pay a higher latency to access the same target model. We believe that guaranteeing both *accuracy parity* and *acceleration parity* across populations is an important and underexplored objective deserving attention.

## 2 RELATED WORK

Speculative decoding has matured from block-wise draft-then-verify ideas into a broad family of methods with theoretical distributional guarantees and practical speed-ups (Leviathan et al., 2023). Subsequent work increases acceptance by improving drafter-target alignment, e.g., knowledge distillation and on-policy data (Zhou et al., 2023), or by enlarging verified structures (token trees) while keeping outputs faithful to the target model (Li et al., 2024). In multilingual inference, specialized drafters can substantially raise acceptance and throughput (Yi et al., 2024a). Parallel to these engineering advances, the multilingual tokenization literature documents large cross-language differences in token counts for semantically equivalent inputs, directly affecting runtime and cost (Petrov et al., 2023a). These efforts show that acceleration is not merely a property of the *algorithm* but of the *algorithm–population match*. Our work is, to our knowledge, the first to (i) model this interaction explicitly as a fairness question about the *distribution of speed-up* across tasks, and (ii) provide a mitigation that promotes equal acceleration across tasks/languages under constraints on faithfulness. (See Appendix B for extended related work).

## 3 BACKGROUND: SPECULATIVE DECODING

Let $q(x|s)$ denote the probability distribution induced by the drafter model $Q_\theta(s)$ over the token $x$ given context $s$, and let $p(x|s)$ represent the corresponding distribution from the verifier model $P_\phi(s)$. We use, $\phi$ and $\theta$ to represent the model parameters, and often suppress the conditioning on $s$ when unambiguous. For a drafted token $x \sim q(x)$, **(1)** if $p(x) \geq q(x)$, the token is immediately accepted. Otherwise, **(2)** the token is rejected with probability $1 - \frac{p(x)}{q(x)}$, and an alternative token is sampled from the residual distribution $p'(x) \triangleq \text{norm}(\max(p(x) - q(x), 0))$. This scheme ensures that generated tokens are sampled from the target distribution $p(x)$ (Leviathan et al., 2023), i.e., without loss in generation quality (see Appendix A.5 for proof).

**Acceptance rate and speed-up.** For a given prefix $s$, the (per-step) acceptance rate[1] $\alpha(s)$ is:

$$\alpha(s) \triangleq \sum_{x \in \mathcal{V}} q(x|s) \min\left(1, \frac{p(x|s)}{q(x|s)}\right) = \sum_{x \in \mathcal{V}} \min\big(q(x|s), p(x|s)\big), \tag{1}$$

where $\mathcal{V}$ is the token vocabulary (shared by $P_\phi$ and $Q_\theta$). We then define $\gamma \in \mathbb{N}$ to be the number of speculative guesses per iteration ($\gamma = 1$ is the single-guess case). Let $c \in [0, 1)$ denote the (platform-dependent) drafter cost ratio, specifically $c = \frac{\text{Drafter Pass Time (s)}}{\text{Verifier Pass Time (s)}}$. Under sufficient parallelism to run the $\gamma + 1$ verifier contexts concurrently[2], the expected wall-time improvement factor (vs. vanilla decoding) for a given $s$ is:

$$\text{Speedup}(s; \gamma, c) = \frac{1 - \alpha(s)^{\gamma+1}}{(1 - \alpha(s))\left[\gamma c + 1\right]}. \tag{2}$$

In particular, for fixed $(\gamma, c)$, Equation (2) is an increasing function of $\alpha(s)$. The next section sheds light on consistent speed-up disparities that emerge during the utilization of speculative inference.

---

[1] Many references denote the one-step acceptance by $\beta_s$; we use $\alpha(s)$ for consistency throughout.

[2] Assuming negligible overhead when $\gamma$ drafted tokens are given to the verifier, then verified in a single pass using parallel inference.

## 4 Unfairness in Speculative Decoding

With the foundation of speculative decoding established, we now showcase the emergence of speed-up disparities across various languages, motivating this study (and further demonstrated in Section 8).

Figure 1 reports the acceptance rates and language-wise benchmark accuracy achieved within speculative decoding on a version of the *multilingual grade-school mathematics* dataset (MGSM) (Shi et al., 2022a), evaluating speed-up across various languages. Notably, both accuracy and acceptance rates vary significantly by language, with low accuracy and low speed-up languages coinciding. We find a 13% gap in average acceptance rates between the fastest and slowest languages, alongside a 52% difference in accuracy. Japanese, the language with the lowest accuracy, also shows the slowest speed-up. Besides high-

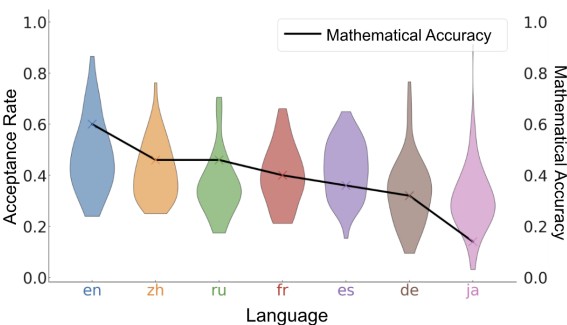

Figure 1: Acceptance rates and accuracies in MGSM using Qwen2.5 series drafter (0.5B) and verifier (3B).

lighting the existence of speed-up disparities, Figure 1 also suggests an important aspect: a connection between language-wise accuracy and speed-up as a driver of unfairness, a connection that this paper explores formally and empirically (see Sections 6 and 8).

## 5 Characterizing Unfairness in Speculative Decoding

We next discuss **(i)** why speed-up is monotone in acceptance, **(ii)** how speed-up is consequently induced by drafter–verifier *fitness*, and **(iii)** how cross-entropy misfit provides an optimizable surrogate for speed-up unfairness with provable implications for acceleration. All proofs are reported in Appendix A.

**Task distributions and task speed-up.** We start by defining a 'task'. Let $\mathcal{V}$ be the shared vocabulary. A *task* $T$ is a distribution over prefixes $s \in \mathcal{V}^*$ (e.g., a language or domain). Consider a finite family $\mathcal{T} = \{T_1, \ldots, T_m\}$; for any task $T \in \mathcal{T}$, define the *task-wise acceptance:*

$$\alpha_T \triangleq \mathbb{E}_{s \sim T}\big[\alpha(s)\big]. \tag{3}$$

Computing the speed-up for a given task proceeds as follows: Firstly, fix the speculative width $\gamma \in \mathbb{N}$ and drafter cost ratio $c \in [0, 1)$ as in Section 3. At a prefix $s$, the expected tokens per verifier iteration is $f_\gamma(\alpha(s)) \triangleq \sum_{k=0}^{\gamma} \alpha(s)^k = \frac{1-\alpha(s)^{\gamma+1}}{1-\alpha(s)}$, (Leviathan et al., 2023), thus the *task-level* speed-up (vs. vanilla) on task $T$ is:

$$S_T \triangleq \mathbb{E}_{s \sim T}[\text{Speedup}(s; \gamma, c)] = \mathbb{E}_{s \sim T}\big[\frac{f_\gamma(\alpha(s))}{1 + \gamma c}\big]. \tag{4}$$

**From divergence to task speed-up.** Next, we highlight a connection between conventional divergence metrics (i.e., cross-entropy, KL-divergence) and task speed-up $S_T$, which will motivate our "unfairness" notion. In particular, the property discussed next makes acceptance the right primitive for analysis and mitigation.

**Theorem 1.** *For $\gamma \geq 1$, the function $f_\gamma : [0, 1) \to \mathbb{R}_+$, defined as $f_\gamma(\alpha) = \sum_{k=0}^{\gamma} \alpha^k$, is increasing on $[0, 1)$ and convex for $\gamma \geq 2$ (linear when $\gamma = 1$). Consequently, for fixed $(\gamma, c)$, the task-level speedup on task T can be lower-bounded by the following divergence functions, resulting in the chain:*

$$S_T \geq \frac{f_\gamma(\alpha_T)}{1 + \gamma c} \geq \frac{f_\gamma\left(1 - \sqrt{\frac{1}{2} \text{KL}_T}\right)}{1 + \gamma c} \geq \frac{f_\gamma\left(1 - \sqrt{\frac{1}{2} \boldsymbol{D}_T}\right)}{1 + \gamma c}, \tag{5}$$

where $\text{KL}_T$ is the task-wise Kullback–Leibler divergence: $\text{KL}_T \triangleq \mathbb{E}_{s \sim T}[D_{\text{KL}}(p(\cdot \mid s) \| q(\cdot \mid s))]$, and $\boldsymbol{D}_T$ is the task-wise cross-entropy: $\boldsymbol{D}_T \triangleq \mathbb{E}_{s \sim T}[-\sum_x p(x|s) \log q(x|s)]$ (proven in Appendix A.1).

The result above highlights three key messages: First, it places $\alpha$ as the singular determinant of speculative speed-up at fixed $(\gamma, c)$, second, the task-level speedup $S_T$ is increasing in $\alpha_T$ (further corollary in Appendix A.2), and third, and most importantly, task-level speed is *monotone* in $D_T$; In particular, the above yields a monotone chain $D_T \downarrow \Rightarrow \alpha_T \uparrow \Rightarrow S_T \uparrow$[3]. This has an important consequence: for fixed $(\gamma, c)$, $D_T$ provides an *optimizable* metric whose reduction *monotonically* tightens a task speed-up. This rationale makes $D_t$ a fitting choice for computing unfairness.

**Unfairness as divergence dispersions.** Next, the section introduces a fairness notion built on the divergence $D_T$, from the rightmost expression of Theorem 1.

**Definition 1.** *(Speculative Decoding Unfairness) For a task family* $\mathcal{T} = \{T_i\}_{i=1}^m$, *speculative decoding unfairness is defined as:*

$$\mathcal{U}(\mathcal{T}) \triangleq \frac{1}{m} \sum_{T \in \mathcal{T}} \left(D_T - D_{\min}\right)^2, \quad where \quad D_{\min} \triangleq \min_{T \in \mathcal{T}} D_T. \tag{6}$$

The larger the quantity $\mathcal{U}(\mathcal{T})$ is, the more disparate the speedups across tasks. Note also that reducing $\mathcal{U}(\mathcal{T})$ contracts the spread of the lower bounds $\{g(D_T)\}_{T \in \mathcal{T}}$, where $g(d) \triangleq f_\gamma\big(1 - \sqrt{\frac{1}{2}d}\big)/(1 + \gamma c)$ is decreasing. Therefore, by Theorem 1, $\downarrow \mathcal{U}(\mathcal{T})$ contracts a certified lower envelope of $\{S_T\}_{T \in \mathcal{T}}$.

## 6 PRECONDITIONS FOR SPEED-UP DISPARITIES

So far, we have established that speed-up disparities appear across tasks, and have formalized *speed-up unfairness* via acceptance (and cross-entropy) misalignment. We now ask: *why do these disparities arise so persistently?* Our thesis is that *disparities in acceleration are primarily driven by disparities in* drafter fitness *across tasks*. We make this precise by relating acceptance to model-task alignment.

**Task misalignment and task fitness.** We first define what is meant by *task-fitness*, and use these notions to reason about the factors that influence task speed-up. For a given task $T$ (a distribution over prefixes $s \in \mathcal{V}^*$), let $u(\cdot \mid s)$ denote the latent task posterior over next tokens (the conditional distribution that generates the data on task $T$). We define *task misalignments* as follows:

$$r_p \triangleq \mathbb{E}_{s \sim T}\Big[\frac{1}{2} \sum_{x \in \mathcal{V}} \big|u(x \mid s) - p(x \mid s)\big|\Big], \qquad r_q \triangleq \mathbb{E}_{s \sim T}\Big[\frac{1}{2} \sum_{x \in \mathcal{V}} \big|u(x \mid s) - q(x \mid s)\big|\Big]. \tag{7}$$

Intuitively, $r_p$ (resp. $r_q$) is the average total variation distance between the verifier's (resp. drafter's) next-token distribution and the task's true posterior (i.e., the expected fraction of probability mass on which the models disagree with $T$ for some prefix, $s$) thus $1 - r$ quantifies 'model→task' alignment. We therefore interpret $1 - r_p$ and $1 - r_q$ as the *task fitness* of the verifier and drafter, respectively. The next result formalizes a relation between this drafter task fitness $(1 - r_q)$ and associated task acceptance.

**Theorem 2** (Drafter-fitness estimator for acceptance). [4] *For any task $T$:*

$$\big|\alpha_T - (1 - r_q)\big| \leq r_p, \tag{8}$$

(proven in A.3). In particular, when the verifier is well-fit to $T$ ($r_p$ small), the acceptance $\alpha_T$ is tightly approximated by the drafter fitness $1 - r_q$. Theorem 2 formalizes the operational intuition: *once the verifier is reasonably aligned with the task, the drafter's task fitness becomes the principal driver of acceptance (and hence acceleration).* Combining Theorems 1 and 2 reveals the monotone chain:

$$\text{drafter fitness } (1 - r_q) \uparrow \Longrightarrow \alpha_T \uparrow \Longrightarrow \text{speed-up}(S_T) \uparrow.$$

The *fairness* implication of this chain is that *given a verifier with high task-fitness, tasks with low drafter fitness will tend to be slower*. In other words, Theorem 2 implies that *disparities in drafter*

---

[3]Throughout, we write $\uparrow x$ ($\downarrow x$) to represent an increase (decrease) to a scalar $x$.

[4]The assumption $r_p \leq r_q$ is typically satisfied: Running speculative decoding in cases where the drafter is more aligned on tasks than the verifier means that generation becomes slower, and of poorer quality relative to vanilla decoding with the drafter alone.

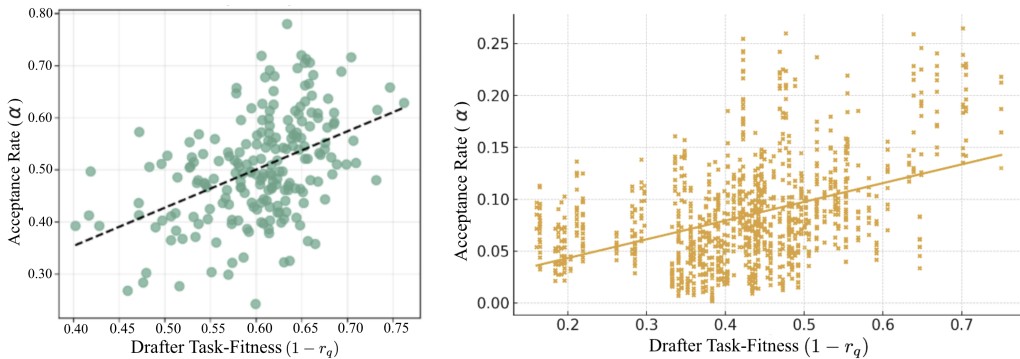

(a) MGSM over Qwen2.5-0.5B and Qwen2.5-3B models

(b) MCoT over Qwen2.5-(0.5B-1.5B), (0.5B-3B), (1.5B-7B), (1.5B-14B), (0.5B-14B) model pairs.

Figure 2: Relation between drafter task-fitness $(1 - r_q)$ and task speed-up / acceptance rates $\alpha$.

*task-fitness have a tendency to produce disparities in speed-up*, with under-fit (and perhaps under-represented) tasks consistently receiving less boost. Accordingly, corollary in Appendix A.4 reveals sufficient conditions for task disparities.

Figure 2 provides empirical evidence from our experiments that evaluate this dependence. It reveals a strong correlation between drafter fitness and speed-up. Firstly, evaluating acceptance rates and drafter fitness over individual examples from the smaller MGSM (Shi et al., 2022b), (with respect to the $P_\phi$ =Qwen2.5-3B, $Q_\theta$ =0.5B model pair), reveals a Spearman coefficient of $r = 0.44$ between drafter fitness and $\alpha$, shown in Figure 2a. We further evaluate the *fitness-speed* relationship on the MCoT (Lai and Nissim, 2024) dataset, leveraging a large set of model pairs ranging in size from 0.5B, to 14B parameters, and aggregating the associated metrics, Figure 2b. We see once again (especially at the extremes), that large drafter task fitness is associated with larger speed-ups and vice versa. These results provide a clear indication that *under-fit tasks are disadvantaged by disparate slowness* (further results in support of this point are provided in Section 8).

## 7 UNFAIRNESS MITIGATION

Motivated by the outlined observations, this section proposes a procedure to reduce speed-up disparities by updating exclusively the *drafter* parameters $\theta$ while keeping the *verifier* $P_\phi$ fixed (to preserve the native decoding behavior of the target model). Indeed, by Theorem 1, lowering $\boldsymbol{D}_T$ increases a certified lower bound on acceptance $\alpha_T$ and hence raises speed-up $S_T$ monotonically. Adjusting $P_\phi$ would compromise exactness and downstream behavior.

**A fairness-weighted descent direction.** To prioritize slow tasks (large $\boldsymbol{D}_T$ while avoiding any incentive to *increase* divergence on the faster tasks, we propose to scale each task's gradient by its *excess divergence*:

$$\widehat{\nabla}_\theta \mathcal{U} \triangleq \frac{1}{m} \sum_{T \in \mathcal{T}} \left( \boldsymbol{D}_T - \boldsymbol{D}_{\min} \right) \nabla_\theta \boldsymbol{D}_T. \tag{9}$$

Equation (9) is exactly the gradient of the objective $\widehat{\mathcal{U}}(\mathcal{T}) = \frac{1}{m} \sum_{T \in \mathcal{T}} (\boldsymbol{D}_T - c)^2$, where $\boldsymbol{D}_{\min}$ is treated as a constant $c$. This has two immediate benefits: **(i)** it pushes down on tasks in proportion to how unfairly slow they are, and **(ii)** the best task ($\boldsymbol{D}_T = \boldsymbol{D}_{\min}$) receives zero weight, so there is *no explicit term* that increases its divergence. Equation (9) performs variance-reduction of $\{\boldsymbol{D}_T\}_{T \in \mathcal{T}}$ around the current floor while still monotonically decreasing the mean divergence.

Now note that, assume a unique minimizer $\boldsymbol{D}_{\min} = \min_T \boldsymbol{D}_T$. Away from ties, differentiating $\mathcal{U}$ yields: $\nabla_\theta \mathcal{U} = \frac{2}{m} \left[ \sum_{T \in \mathcal{T}} (\boldsymbol{D}_T - \boldsymbol{D}_{\min}) \nabla_\theta \boldsymbol{D}_T - \sum_{T \in \mathcal{T}} (\boldsymbol{D}_T - \boldsymbol{D}_{\min}) \nabla_\theta \boldsymbol{D}_{\min} \right]$. The second term (blue colored) is problematic: it can *intentionally* move $\boldsymbol{D}_{\min}$ upward to reduce dispersion, thereby degrading speed-up on the best task. The gradient proposed in Equation (9) removes this term and thus avoids any *direct* incentive to harm $\boldsymbol{D}_{\min}$.

**Stochastic corrective drafter fine-tuning (s-CDF).** In practice, Equation 9 is applied by estimating $\boldsymbol{D}_T$ and $\nabla_\theta \boldsymbol{D}_T$ from mini-batches. For a batch $\mathcal{B}_T \subset T$:

$$\widehat{\boldsymbol{D}}_T = \frac{1}{|\mathcal{B}_T|} \sum_{s \in \mathcal{B}_T} \Big[ -\sum_x p(x \mid s) \log q_\theta(x \mid s) \Big], \quad \nabla_\theta \widehat{\boldsymbol{D}}_T = -\mathbb{E}_{s \in \mathcal{B}_T, \, x \sim p(\cdot|s)} \big[ \nabla_\theta \log q_\theta(x \mid s) \big].$$

(10)

We adopt this batched approach for temporal efficiency, enabling fast computation of task-wise disparities, and then apply the gradient in Equation 9 per step resulting in the corrective finetuning process defined in Algorithm 1.

## 8 EXPERIMENTAL ANALYSIS

In this section, building upon and extending the theoretical insights and fairness objectives discussed, we present key findings from our empirical analysis. The analysis will highlight that: **(1)** the computational speed-up gained from speculative decoding is not uniformly distributed, (validated across multiple model pairs and datasets). **(2)** Languages that receive disproportionate speed-up tend to be underrepresented within conventional training corpus's. **(3)** Stochastic corrective drafter fine-tuning serves as an effective speed-up unfairness mitigation across multiple models.

---

**Algorithm 1:** Stochastic Corrective Drafter Fine-tuning (s-CDF)

**Input:** $\{T_1, \ldots, T_m\}$; verifier $P_\phi$ (frozen); drafter $Q_\theta$; optimizer $A$; batch sizes $\{\mathcal{B}_T\}_{T \in \mathcal{T}}$; step size $\beta$.

**while** *not converged* **do**

> Sample mini-batches $\mathcal{B}_T \subset T$ of size $\mathcal{B}_T$ for all $T \in \mathcal{T}$.
> Estimate $\widehat{\boldsymbol{D}}_T$, $\nabla_\theta \widehat{\boldsymbol{D}}_T$ on each $\mathcal{B}_T$.
> Set $\boldsymbol{D}_{\min} \leftarrow \min_T \widehat{\boldsymbol{D}}_T$ and $c \leftarrow \widehat{\boldsymbol{D}}_{\min}$.
> $\Delta_\theta \leftarrow -\frac{1}{m} \sum_{T \in \mathcal{T}} (\widehat{\boldsymbol{D}}_T - c) \nabla_\theta \widehat{\boldsymbol{D}}_T$.
> $\theta \leftarrow A(\theta, \beta, \Delta_\theta)$.

---

### 8.1 DATASETS AND MODELS

We model each task as a distinct linguistic group and conduct experiments over seven model pairs from the Qwen2.5 family, and four multilingual datasets, spanning bilingual pretraining, multilingual open-ended generations and multilingual mathematics:

• **Bilingual Web-Text:** We use English (L1) and Japanese (L2) web text corpora as seed-text, as over- and under-represented languages, respectively. The English dataset is sourced from a small web-text corpus (nampdn-ai, 2023), and Japanese data is drawn from the Japanese WikiNews and related sources curated in (fujiki, 2023).

• **Multilingual Dolly (Üstün et al., 2024):** This dataset is drawn from the Aya Evaluation Suite (Singh et al., 2024) containing **open-ended instruction-following** examples. We use its machine-translated version, which includes 200 aligned prompt-response pairs per language. Languages are selected from the set officially supported by the Qwen2.5 series, and their relative representation is estimated using internal linguistic prior probabilities[5]. We use BLEU (Papineni et al., 2002) as a reference-based metric for assessing the fidelity of our model outputs relative to reference solutions.

• **MGSM (Multilingual Grade-School Math) (Shi et al., 2022b; Lai and Nissim, 2024):** This benchmark consists of **grade-school mathematical problems**. It is used to examine the relations between task accuracy and speed-up ($\alpha$), as well as to evaluate the efficacy of our s-CDF method. Given the sample intensive nature of finetuning we use the larger MCoT (Lai and Nissim, 2024) version of the MGSM for s-CDF experiments, as well as the smaller MGSM (Shi et al., 2022b) for evaluation-heavy experiments. The benchmark provides reference solutions that can be used to verify model accuracy in a discrete manner.

Additionally, we evaluate seven pairs of models in total. For the bilingual experiments, we use **GPT-2** (117M) (OpenAI, 2019a) as the drafter and **GPT-2-XL** (1.5B) (OpenAI, 2019b) as the verifier, both of which were pretrained on a private web-text based corpus. For Multilingual Dolly we focus on the **Qwen2.5-0.5B** and **Qwen2.5-3B** pair. Similarly for MGSM, and s-CDF experiments we use five pairs from the **Qwen-2.5** family, with sizes ranging from 0.5B to 14B parameters.[6]

### 8.2 DISPARATE SPEED-UPS AND UNDERREPRESENTED LANGUAGES

We now extend the results presented in Section 4, showing that disparities in speculative decoding speed-ups are consistent, as well as provide evidence in favor of relationships between language "representation" and speed-up.

---

[5]We estimate representation based on the prior distribution of drafter/verifier models, detailed in Section 8.2.

[6]We opt for Qwen2.5 due to their broad multilingual support and the variety of model sizes that are offered.

**Bilingual disparities.** We begin by comparing speculative decoding performance between English and Japanese. As shown in Figure 3, the acceptance rate $\alpha$ is consistently higher for English than for Japanese (62.5% vs. 54.5%). This mirrors the misalignment levels between drafter and verifier on each language, with divergence $D(\cdot)$ measured at 0.47 for English and 1.08 for Japanese. This bilingual setup highlights clear speed-up disparity.

**Multilingual grade-school math (MGSM).** We further evaluate this phenomenon in the context of mathematical reasoning using the MGSM benchmark. For each supported language, speculative decoding is performed over chain-of-thought style problem-solving prompts. Final answers are evaluated via pattern matching to compute task accuracy, and speculative decoding is run with Qwen2.5-

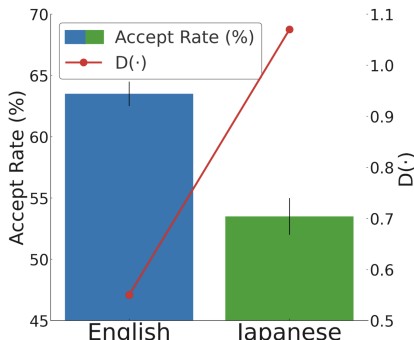

Figure 3: Divergence $D(\cdot)$ is high (low) for slower (faster) language.

0.5B, Qwen2.5-3B, for drafter and verifier respectively. Ultimately, Figure 4 reveals a strong positive correlation between per-language accuracy and acceptance rate $\alpha$, indicating that *languages benefiting from higher fitness (higher resource languages) also benefit from lower latency during decoding*.

We also explore the larger MGSM benchmark, (MCoT) (Lai and Nissim, 2024), featuring an extensive number of languages and mathematical questions, including rare, low-resource languages such as Bangla and Telugu. We utilize the same model pair; drafter (Qwen2.5-0.5B) and verifier (Qwen2.5-3B), then evaluate acceptances rates over problems on each language, reported in Figure 5. We continue to see large disparities of up to 65% between our fastest and slowest languages (English vs Japanese), with languages like Telugu experiencing 60% lower speed-up relative to English, revealing that *large speed-up disparities persist across variations in dataset*, as well as across different groups of languages.

**Multilingual DollyQA.** Next, we evaluate $\alpha$ across a diverse set of languages supported by both DollyQA and Qwen 2.5. For each language $L$, we randomly sample prompts and compute the average acceptance rate $\alpha_L$ under speculative decoding.

To investigate the relationship between $\alpha_L$ and language representation, we first sample $K$ generations from our drafter on an empty prefix $\emptyset$[7], resulting in the collection of generations: $X = (s_1 \sim Q_\theta(\emptyset), \ldots, s_K \sim Q_\theta(\emptyset))$. We then confirm the language of each sample with an operator $F(s)$[8], resulting in a set of corresponding languages $L = (l_1 = F(s_1), \ldots, l_K = F(s_K))$. We then estimate the vector $P$, where $P_i = p(l_i)$ denotes the probability of sampling from language $l_i$ for $N$ valid languages. We

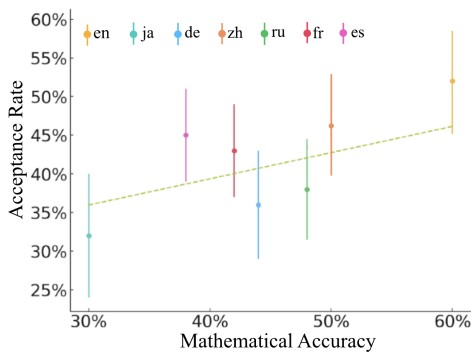

Figure 4: Alpha against task accuracy within different languages on MGSM data.

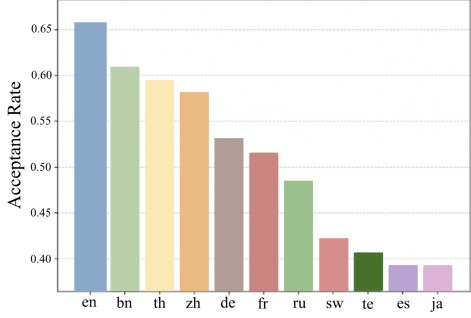

Figure 5: Acceptance rates for Qwen2.5-0.5B, 3B model pair, on larger MCoT dataset.

compute this estimate with $P \approx \hat{P} = \frac{1}{K}(\Sigma_{s_j \sim X}\mathbb{I}[(F(s_j) = l_1], \ldots, \Sigma_{s_j \sim X}\mathbb{I}[(F(s_j) = l_N])$[9]. The resulting probabilities, $\hat{P}$ represent model priors with respect to our languages. We finally rank our languages by their estimated representation probability from 'most' to 'least' represented.

The resulting trend is clear. As illustrated in Figure 6 (next page), we observe a strong inverse correlation between language rank and $\alpha$, indicating that less represented languages tend to exhibit

---

[7]Specifically, we pass the empty string ' ' to our models.
[8]Using the NLTK library to classify languages (Loper and Bird, 2002).
[9]Approximation is shown due to finite sampling error. In practice we set a large $K$ of 100,000.

lower speed-ups, further demonstrating the disproportionate speed-up benefit high-resource languages receive during speculative-decoding.

## 8.3 MITIGATING SOLUTIONS

Next, we assess multiple strategies for reducing disparities in speculative decoding, establishing s-CDF as a promising mitigation method.

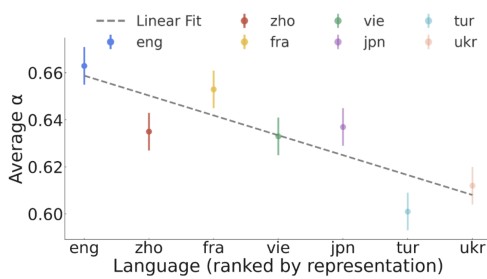

**Joint temperature optimization.** When verifier and drafter temperatures move in tandem, higher temperatures bring distributions toward uniformity, increasing speed-up but degrading generation quality. Similarly, one-hot distributions at temperature zero may lead to higher acceptances, however, this set-up suffers from the same quality degradation, especially in smaller verifiers (Nakaishi et al., 2024). Experiments on DollyQA show that $\alpha$ thus follows

Figure 6: Expected speed up for each language sorted by *representation rank*.

a parabolic trend over temperature, with minima around $T \approx 1.5$ (Figure 7a). However, quality-adjusted speed-up—computed as $\alpha \cdot \beta$, where $\beta$ is BLEU—remains somewhat flat across temperature settings (Figure 7b). This suggests that while temperature affects $\alpha$, it does so in a way that undermines output quality, offering no practical fairness benefit. We accordingly focus our attention to methods that do not influence verifier generations.

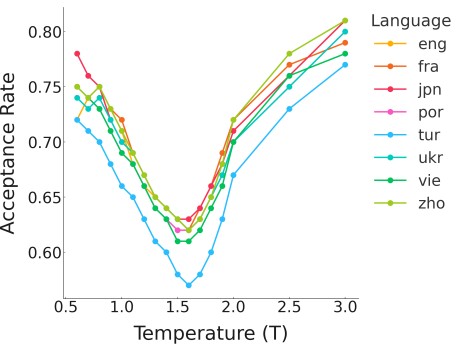

(a) Alpha at different temperatures for each language

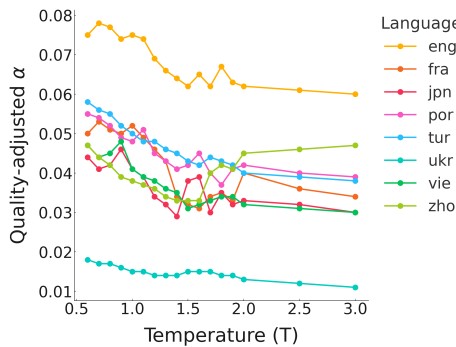

(b) Quality adjusted alpha at different temperatures for each language.

Figure 7: Degrading influence of temperature over DollyQA data. Consistent parabolic alpha distributions, roughly flat quality-adjusted alpha distributions.

**Data balancing.** We next investigate the influence that differing data proportions during finetuning have on speed-up fairness, exploring a mixture-based finetuning strategy in the bilingual (English-Japanese) setting. Varying the proportion of Japanese data shows that increasing its prevalence improves Japanese drafter loss and acceptance rates, while inducing minor regressions in English performance (Figures 8a, 8b). Unfairness, $\mathcal{U}(\cdot)$, decreases as Japanese data increases, showing that basic data re-balancing can have beneficial effects, as well as shows the effects data representation has on speed-ups (Figure 8c). However, this approach lacks principled guidance for setting data ratios, and is not clearly scalable to several tasks, prompting further exploration.

**Stochastic corrective drafter finetuning (s-CDF).** Recall our proposed unfairness function $\mathcal{U}$. We utilize our derived projected gradient, Equation 9, to take descent steps along our function, $\mathcal{U}$. We find that for a set of languages, optimizing the objective $\mathcal{U}$ stochastically, is the most practical, while still leading to unfairness convergence. In these experiments, we use a batch size of $\mathcal{B} = 8$, and select five languages from our MCoT dataset that show initial unfairness (English, Spanish, Russian, Mandarin, German). We repeat experiments over multiple model pairs from the Qwen2.5 series, testing across five model pairs with 0.5B, and 1.5B parameter models as drafters, and 3B, 7B, and 14B parameter models as verifiers. We see on average, a *20% reduction* in the variance of our acceptance rates across our model pairs during finetuning, alongside, a *12% decrease* in our unfairness $\mathcal{U}$. The mutual

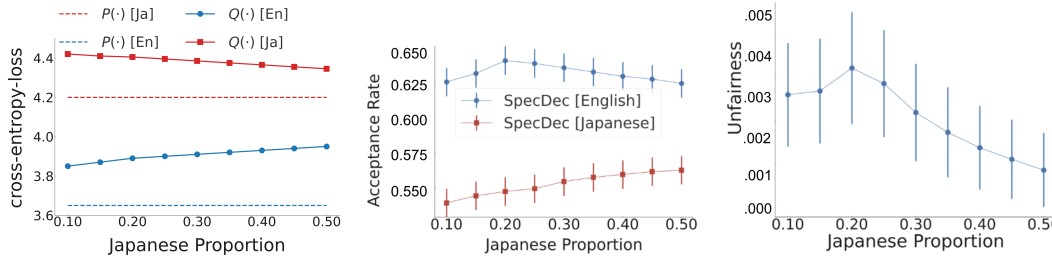

(a) $P$ and $Q$ losses on 'En' and 'Ja' during progressive finetuning.

(b) Language acceptance rates during data progressive finetuning.

(c) Unfairness objective during data progressive finetuning.

Figure 8: Data progressive finetuning in bilingual setting.

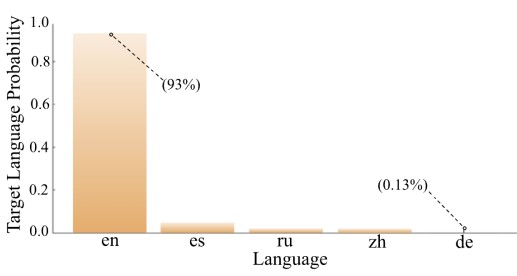

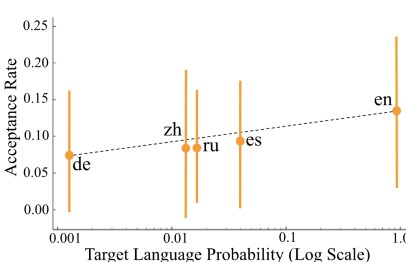

(a) Target language probability for each language during finetuning, with 'true' target language (en) highlighted.

(b) Acceptance rate correlation with target probability

Figure 9: Language sampling probabilities and acceptance rates during the proposed corrective drafter tuning (s-CDF)

reduction in speed-up variance and speed-up unfairness $\mathcal{U}$ serves to showcase the connection between our unfairness metric and speed-up dispersions empirically, as was established in Theorem 1.

We also see that our fastest language, English, is sampled as our target language in 93% of cases, Figure 9a, in other words giving English a 'target language probability' of 93%, while our slowest language, German, features gets sampled as $D_{\min}$ 0.13% of batches, Figure 9a. This showcases that our stochastic approach tends to preserve information about speed-up disparities even with small batch sizes ($\mathcal{B} = 8$), as well as speaks to the persistent speed-up unfairness present in multilingual datasets. Subsequently, when studying the relationship between the target language probability and the acceptance rates, we observe a positive trend between the target probability for a language, and the acceptance rates for the language $\alpha$, see 9b, showcasing the predictive power of our divergence metric $D(\cdot)$ at forecasting speed-ups even in stochastic contexts.

## 8.4 ABLATION STUDIES

Finally, the paper details empirical results to further support the previous decisions. Among other aspects, these results highlight the empirical consequences of the 'degrading gradient term' (see Section 5), as well as the advantages of s-CDF over prior distillation techniques.

For mitigation ablations, we compare **(i)** the s-CDF algorithm (Algorithm 1), **(ii)** the s-CDF algorithm with the 'Degrading Term' included (see Section 7), **(iii)** distillation with uniform weights across tasks (DistillSpec from Zhou et al. (2024)) and **(iv)**, s-CDF where cross-entropy is replaced with TV loss. The mitigation's are run over six languages (English, German, Spanish, Japanese, Russian Swedish, Mandarin), and across several model pairs from the Qwen2.5 family, ranging from 1.5B to 32B verifiers. The full results for 1.5B and 3.0B verifiers are detailed in Table 1 (see Appendix C for other models). The training is run for a fixed number of steps for all methods, with identical data exposure ($\sim 1M$ tokens). We report average change in acceptance across languages ($\Delta \text{Mean}(\alpha_T)$), change in fastest language ($\Delta \alpha_{\max}$, generally 'English'), and unfairness reduction ($\Delta \mathcal{U}$).

Table 1: Finetuning ablations for Qwen2.5-0.5B $\rightarrow$ {1.5B, 3B} showing changes in average acceptance, fastest-language acceptance, and unfairness.

| Qwen2.5-0.5B $\rightarrow$ 1.5B | | |
| --- | --- | --- |
| **Method** | $\uparrow \%\Delta\mathrm{Mean}(\alpha_T)$ | $\uparrow \%\Delta\alpha_{\max}$ | $\downarrow \%\Delta\mathcal{U}$ |
| **s-CDF (ours)** | $+5.3\%$ | $+3.2\%$ | $-\mathbf{76.7}\% \pm 0.93$ |
| s-CDF (w/ Degrading Term) | $-0.1\%$ | $-0.7\%$ | $-19.4\% \pm 1.62$ |
| DistillSpec | $+\mathbf{5.8}\%$ | $+\mathbf{4.1}\%$ | $-64.0\% \pm 2.15$ |
| s-CDF (w/ TV Loss) | $+4.4\%$ | $+1.8\%$ | $-30.3\% \pm 1.36$ |
| **Qwen2.5-0.5B $\rightarrow$ 3B** | | |
| **Method** | $\uparrow \%\Delta\mathrm{Mean}(\alpha_T)$ | $\uparrow \%\Delta\alpha_{\max}$ | $\downarrow \%\Delta\mathcal{U}$ |
| **s-CDF (ours)** | $+5.1\%$ | $+2.91\%$ | $-\mathbf{78.1}\% \pm 0.63$ |
| s-CDF (w/ Degrading Term) | $-0.3\%$ | $-0.8\%$ | $-12.9\% \pm 1.28$ |
| DistillSpec | $+\mathbf{5.7}\%$ | $+\mathbf{3.3}\%$ | $-62.4\% \pm 2.21$ |
| s-CDF (w/ TV Loss) | $+4.2\%$ | $+1.5\%$ | $-29.8\% \pm 1.45$ |

The results are largely consistent with the paper's discussion. As expected, we see that including the degrading term $-(D_T - D_{\min})\nabla_\theta D_T$ causes a slight reduction in the speed-up of the fastest language (English; $-0.7\%$ and $-x\%$, Table 1), as well as leads to less unfairness convergence than s-CDF. We argue that gradient ascent over high-quality data (on account of the degrading term) introduces instability that worsens unfairness convergence (yet this is left open for future work). Secondly, we see that uniform DistillSpec results in large mean and fastest-lang increases, as expected, yet produces **sub-optimal unfairness-mitigation**, exhibiting lower mitigation magnitude than s-CDF. This is because minority and majority languages are equally weighted in the DistillSpec objective, with no notion of disparities. Finally, we observe that using TV loss in place of CE-Loss ('w/ TV Loss' above) results in slower convergence across all metrics (TV: $-30.3\%$ and $-29.8\%$ vs s-CDF: $-76.7\%$ and $-78.1\%$ unfairness reduction), and thus lower unfairness mitigation.

Generally, these ablations justify some of the decisions made throughout the paper (i.e., using CE instead of TV, removing the Degrading-Term) and highlight the value of a fairness sensitive optimization strategy relative to previous approaches.

## 9 CONCLUSION

This work reveals a previously overlooked source of unfairness in accelerated inference: speculative decoding yields unequal speed-up benefits across tasks and languages. We find that underrepresented or under-fit distributions—such as low-resource languages consistently receive lower speed-up motivated by disparities in drafter fitness. To understand and mitigate this disparity, we conducted a comprehensive empirical study across multilingual benchmarks, and show the consistency of this fairness issue. This analysis was driven by theoretical intuitions presented in our work, showing a connection between task-wise acceptance and the drafter task fitness. Finally, we proposed a mitigation technique based on a projected gradient-based method that reduces speed-up disparities by selectively improving under-performing tasks. We believe these results are important as they draw attention to a potential source of computational inequity and that guaranteeing both accuracy parity and acceleration parity across populations is an area deserving attention.

## ETHICS STATEMENT

Our work concerns the ethical deployment of speculative decoding in a manner that is inherently fairness-aware with respect to various, potentially discriminated subgroups. Our analysis cautions practitioners against ignoring the disparate effects that inference acceleration algorithms can have on certain subgroups. However, the warnings communicated in this paper, strictly speaking, do not prevent the misuse or irresponsible usage of speculative decoding yet the emphasis on mitigation techniques, we believe, contributes to the open problem of fair, responsible and ethical AI usage.

## REPRODUCIBILITY STATEMENT

We documented all artifacts required to reproduce our results in the main paper and Appendix: checkpoints for speculative decoding (draft/verify), our fairness metric $\mathcal{U}$ (cross-entropy–based), logging utilities (acceptance rate, tokens-per-step, realized speed-up), and s-CDF algorithms. Upon release, our repository includes exact *model identifiers*, tokenizer versions, acceptance criteria, and hyperparameters; configuration files (YAML) enumerate drafter–verifier pairs, batch sizes, maximum lengths, and stopping rules. We fix and document *random seeds* (for data sampling and any stochastic training), report *hardware* (GPU model, driver/CUDA versions) and key libraries used, and provide command lines to reproduce every table/figure. For datasets, we provide citation instructions and will publish *frozen evaluation lists* (document IDs and prompts) to avoid data drift. Finally, we also plan to report wall-clock compute and carbon estimates for major runs.

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
