# A    MISSING PROOFS

## A.1    RELATING DIVERGENCE AND SPEED-UP VIA A MONOTONE CHAIN

**Theorem 1.** *For $\gamma \geq 1$, $f_\gamma : [0,1) \to \mathbb{R}_+$ is strictly increasing (and convex for $\gamma \geq 2$). Consequently, for fixed $(\gamma, c)$, there is a monotone chain from $S_T$ to $\boldsymbol{D}_T$:*

$$S_T \;\geq\; \frac{f_\gamma\Big(1 - \sqrt{\frac{1}{2}\,\mathrm{KL}_T}\Big)}{\gamma c + 1} \;\geq\; \frac{f_\gamma\Big(1 - \sqrt{\frac{1}{2}\,\boldsymbol{D}_T}\Big)}{\gamma c + 1}\,, \tag{11}$$

*where $T$ is a task-distribution over prefixes, $f_\gamma(\alpha) = \frac{1-\alpha^{\gamma+1}}{1-\alpha}$ is expected accepted tokens, and $S_T$ is the task-wise speed-up, defined as $S_T = \frac{f_\gamma(\alpha(s))}{\gamma c + 1}$. Finally, task-wise Kullback–Leibler and cross-entropy are $\mathrm{KL}_T = \mathbb{E}_{s \sim T}[\mathrm{KL}(p\|q)]$ and $\boldsymbol{D}_T = \mathbb{E}_{s \sim T}[H(p,q)]$ with $H(p,q) = \mathbb{E}_{x \sim p}[-\log q(x)]$, for verifier, drafter posteriors $p(x), q(x)$ respectively on next-token $x$.*

*Proof.* First, $f_\gamma'(\alpha) = \sum_{k=1}^\gamma k\,\alpha^{k-1} > 0$ for $\alpha \in [0,1)$, so $f_\gamma$ is strictly increasing; and $f_\gamma''(\alpha) = \sum_{k=2}^\gamma k(k-1)\alpha^{k-2} \geq 0$, with strict convexity for $\gamma \geq 2$ on $(0,1)$.

By Jensen on convex $f_\gamma$ ($\gamma \geq 2$; equality when $\gamma = 1$ as $f_\gamma$ is affine):

$$\mathbb{E}[f_\gamma(\alpha(s))] \;\geq\; f_\gamma(\mathbb{E}[\alpha(s)]) \;=\; f_\gamma(\alpha_T).$$

Divide by $\gamma c + 1$ to get the bound for $S_T$.

By Pinsker, $\mathrm{TV}(p,q) \leq \sqrt{\frac{1}{2}\mathrm{KL}(p\|q)}$, hence $\alpha(s) = 1 - \mathrm{TV}(p,q) \geq 1 - \sqrt{\frac{1}{2}\mathrm{KL}(p\|q)}$. Taking expectations and concavity of the square root yields $\alpha_T \geq 1 - \sqrt{\frac{1}{2}\,\mathrm{KL}_T}$. Using $D_T = H(p) + \mathrm{KL}_T \geq \mathrm{KL}_T$ implies the second inequality. Monotonicity of $f_\gamma$ finishes the chain. $\qquad\square$

## A.2    SPEED-UP AND ACCEPTANCE DEPENDENCY

**Corollary 1.** *For $\gamma \geq 1$, $f_\gamma : [0,1) \to \mathbb{R}_+$ strictly increasing, we conclude that $\mathrm{Speedup}(s; \gamma, c)$ is strictly increasing in $\alpha(s)$.*

*Proof.* $f_\gamma'(\alpha) = \sum_{k=1}^\gamma k\,\alpha^{k-1} > 0$ for $\alpha \in [0,1)$, so $f_\gamma$ is strictly increasing, as stated, with strict convexity for $\gamma \geq 2$ on $(0,1)$. Therefore the claim follows since $\mathrm{Speedup}(s; \gamma, c) = \frac{f_\gamma(\alpha(s))}{\gamma c + 1}$ shares the monotonicity of $f_\gamma$. $\qquad\square$

## A.3    RELATING DRAFTER-FITNESS AND ACCEPTANCE

**Theorem 2.** *Let $u(\cdot \mid s)$ be the latent task posterior. Define task misfits*

$$r_p \;\triangleq\; \mathbb{E}_{s \sim T}\Big[\frac{1}{2}\sum_{x \in V}\big|u(x \mid s) - p(x \mid s)\big|\Big], \qquad r_q \;\triangleq\; \mathbb{E}_{s \sim T}\Big[\frac{1}{2}\sum_{x \in V}\big|u(x \mid s) - q(x \mid s)\big|\Big].$$

*Then*

$$\big|\alpha_T - (1 - r_q)\big| \;\leq\; r_p.$$

*Proof.* By triangle inequality and its reverse for total variation at each prefix $s$:

$$\big|\,\mathrm{TV}(p,q) - \mathrm{TV}(u,q)\,\big| \;\leq\; \mathrm{TV}(u,p) \;\Rightarrow\; \big|(1 - \alpha(s)) - (r_q(s))\big| \;\leq\; r_p(s),$$

where $r_q(s) = \mathrm{TV}(u,q)$ and $r_p(s) = \mathrm{TV}(u,p)$. Averaging over $s \sim T$ and using Jensen on $\big|(1 - \alpha(s)) - (r_q(s))\big| \leq r_p(s)$ yields $\big|(1 - \alpha_T) - r_q\big| \leq r_p$, i.e., $\big|\alpha_T - (1 - r_q)\big| \leq r_p$. $\qquad\square$

## A.4 A SUFFICIENT CONDITION FOR DISPARITIES

**Corollary 2.** *For two tasks $T_i, T_j$ with pairs $(r_p^i, r_q^i)$ and $(r_p^j, r_q^j)$, a strict acceptance gap $\alpha_{T_i} > \alpha_{T_j}$ is guaranteed whenever*

$$r_q^j - r_q^i \; > \; r_p^i + r_p^j.$$

*Moreover, by Corollary 1.,*

$$S_{T_i} - S_{T_j} \; \geq \; \frac{\alpha_{T_i} - \alpha_{T_j}}{\gamma c + 1} \; > \; 0.$$

*Proof.* From Theorem 2, $\alpha_{T_i} \geq 1 - (r_q^i + r_p^i)$ and $\alpha_{T_j} \leq 1 - (r_q^j - r_p^j)$ (since $r_q^j \geq r_p^j$ by assumption). The stated condition implies $1 - (r_q^i + r_p^i) > 1 - (r_q^j - r_p^j)$, hence $\alpha_{T_i} > \alpha_{T_j}$. Apply Corollary 1. $\square$

## A.5 CORRECTNESS OF SPECULATIVE SAMPLING

**Theorem 3.** *Tokens sampled via* speculative sampling *from $p(x)$ and $q(x)$ are distributed identically to those sampled from $p(x)$.*

*Proof.* (As reported in **?**) Let $\alpha$ be the acceptance probability. Note that as $p'(x) = \text{norm}(\max(0, p(x) - q(x))) = \frac{p(x) - \min(q(x), p(x))}{\sum_{x'}(p(x') - \min(q(x'), p(x')))} = \frac{p(x) - \min(q(x), p(x))}{1 - \alpha}$, the normalizing constant for the adjusted distribution $p'(x)$ is $1 - \alpha$.

Now:

$$P(x = x') = P(\text{guess accepted}, x = x') + P(\text{guess rejected}, x = x')$$

Where:

$$P(\text{guess accepted}, x = x') = q(x') \min(1, \frac{p(x')}{q(x')}) = \min(q(x'), p(x')) \tag{12}$$

$$P(\text{guess rejected}, x = x') = (1 - \alpha)p'(x') = p(x') - \min(q(x'), p(x')) \tag{13}$$

And thus, overall we obtained the sought result:

$$P(x = x') = \min(p(x'), q(x')) + p(x') - \min(p(x'), q(x')) = p(x'). \tag{14}$$

$\square$

## A.6 TASK OVERLAP

We formalize the intuition that "the more similar two tasks are, the more similar their divergences are." Let

$$p_\cap(T_1, T_2) \; := \; 1 - TV(T_1(s), T_2(s)),$$

be the *overlap coefficient* between $T_1$ and $T_2$, which satisfies $0 \leq p_\cap \leq 1$. Define the disparity

$$\delta \; = \; \frac{1}{2}|\boldsymbol{D}_{T_1} - \boldsymbol{D}_{T_2}|, \qquad \boldsymbol{D}_T \; = \; \mathbb{E}_{s \sim T}[CE(p(\cdot|s), q(\cdot|s))].$$

**Theorem 4.** *If $p_\cap$ increases while the conditional expectations of $\boldsymbol{D}(\cdot)$ on the non-overlapping regions of $T_1$ and $T_2$ remain fixed, then $\delta$ is strictly decreasing in $p_\cap$, i.e.*

$$\frac{d}{dp_\cap}\delta(p_\cap) < 0.$$

*Proof.* Write each task as a decomposition into overlapping and non-overlapping sets:

$$T_1 = p_\cap T_\cap + (1 - p_\cap) T_1 \setminus T_2, \qquad T_2 = p_\cap T_\cap + (1 - p_\cap) T_2 \setminus T_1,$$

where $T_\cap$ denotes the normalized restriction of both tasks to the region where $T_1$ and $T_2$ overlap.

By linearity of expectation,

$$\boldsymbol{D}_{T_1} = p_\cap \mathbb{E}_{T_\cap}[\boldsymbol{D}] + (1 - p_\cap) \mathbb{E}_{T_1 \setminus T_2}[\boldsymbol{D}],$$

$$\boldsymbol{D}_{T_2} = p_\cap \, \mathbb{E}_{T_\cap}[\boldsymbol{D}] + (1 - p_\cap) \, \mathbb{E}_{T_2 \setminus T_1}[\boldsymbol{D}].$$

Subtracting gives

$$\boldsymbol{D}_{T_1} - \boldsymbol{D}_{T_2} = (1 - p_\cap) \left( \mathbb{E}_{T_1 \setminus T_2}[\boldsymbol{D}] - \mathbb{E}_{T_2 \setminus T_1}[\boldsymbol{D}] \right).$$

Hence the disparity is

$$\delta(p_\cap) = \frac{1}{2} \left| (1 - p_\cap) \, \Delta \right|, \qquad \Delta := \mathbb{E}_{T_1 \setminus T_2}[\boldsymbol{D}] - \mathbb{E}_{T_2 \setminus T_1}[\boldsymbol{D}].$$

Since $|\Delta|$ is constant in $p_\cap$,

$$\frac{d}{dp_\cap} \delta(p_\cap) = -\frac{1}{2} \, |\Delta| < 0.$$

Thus increasing task overlap $p_\cap$ strictly decreases task divergence disparity $\delta$. $\qquad\square$

### A.7   TASK-WISE MISALIGNMENT LEVELS

**Theorem 5.** *For models $P, Q$ with task misalignment levels $r_{q_T}$ and $r_{p_T}$ on task $T$, the acceptance probability $\alpha_T$ is bounded by*

$$\alpha_T \geq 1 - \left( r_{q_T} + r_{p_T} \right).$$

*Proof.* By definition, the acceptance probability on task $T$ is

$$\alpha_T = 1 - \tfrac{1}{2} \, \mathbb{E}_{x \sim T}[\| p(\cdot \mid x) - q(\cdot \mid x) \|_1].$$

Since $r_{p_T}$ and $r_{q_T}$ are total variation misfits with respect to the latent task distribution $u(\cdot \mid x)$, they satisfy

$$r_{p_T} = \mathbb{E}_{x \sim T}\left[ \tfrac{1}{2} \| u(\cdot \mid x) - p(\cdot \mid x) \|_1 \right], \qquad r_{q_T} = \mathbb{E}_{x \sim T}\left[ \tfrac{1}{2} \| u(\cdot \mid x) - q(\cdot \mid x) \|_1 \right].$$

Applying the triangle inequality at each prefix $x$ gives

$$\| p(\cdot \mid x) - q(\cdot \mid x) \|_1 \leq \| p(\cdot \mid x) - u(\cdot \mid x) \|_1 + \| u(\cdot \mid x) - q(\cdot \mid x) \|_1.$$

Taking expectations over $x \sim T$ and noting that each term $\| u - p \|_1$ and $\| u - q \|_1$ contributes a factor of 2 when expressed in terms of total variation, we obtain

$$\mathbb{E}_{x \sim T}[\| p(\cdot \mid x) - q(\cdot \mid x) \|_1] \leq 2(r_{p_T} + r_{q_T}).$$

Substituting this into the acceptance definition yields

$$\alpha_T \geq 1 - \tfrac{1}{2} \cdot 2(r_{p_T} + r_{q_T}) = 1 - (r_{p_T} + r_{q_T}).$$

This proves the claim. $\qquad\square$

## B   EXTENDED RELATED WORK

**Fairness in LLMs.** With the widespread use of Large Language Models (LLMs), concerns regarding fairness have become increasingly prominent. Notably, fairness-related issues in LLMs can manifest in the form of toxicity in generated outputs and biases that cause harm to various social groups. In particular, a well-established distinction (Chu et al., 2024, Gallegos et al., 2023) exists between *representational harms*—such as the use of derogatory language, disparate system performance, erasure, misrepresentation, and stereotyping, which contribute to denigrating and subordinating attitudes toward certain social groups—and *allocational harms*, which involve the amplification of existing biases, the creation of new biases, and the reinforcement of stereotypes.

While significant efforts have been made to address these issues through various fairness metrics (Dhamala et al., 2021; Delobelle et al., 2022, Czarnowska et al., 2021) and mitigation techniques (Steed et al., 2022; Omrani et al., 2023, Kumar et al., 2023), the problem remains far from solved, as biases can be introduced at multiple stages, including through training prompts, labeling choices, or at the embedding level.

For a comprehensive overview of fairness issues in LLMs, we refer the reader to the survey by Gallegos et al. (2023) and the taxonomic survey by Chu et al. (2024).

**Multilingual LLMs.** In this work, we focus on the issue of disparate system performance across different populations, where these populations are represented by different languages. Specifically, we examine fairness across languages in the context of *speculative decoding*, where disparities may arise due to varying time and cost requirements for generating outputs in different languages.

The problem of fairness in multilingual LLMs has been studied from various perspectives. For instance, Petrov et al. (2023a) highlight how language model tokenizers introduce unfairness between languages, while Yi et al. (2024a) demonstrate that language-specific draft models, optimized through a targeted pretrain-and-finetune strategy, significantly improve inference speed compared to previous methods.

Importantly, while the study of the impact of techniques to speed up and reduce the impact of using LLMs has already been investigated (Das et al., 2024), our study combines speculative decoding fairness and multilingual LLMss. In addition, we opt to offer detailed explanation for these effects and rigorous mitigation strategies in a manner that is distinct from previous works.

**Speculative decoding.** Where prior works like Yi et al. (2024b) examine the deployment of speculative decoding in multilingual settings, evaluating over multiple sub-tasks, they ignore critical fairness questions which we address. Additionally, works like Zhou et al. (2023) discuss different notions of distributional misalignment, which they use to optimize speed-up on tasks. However, they do not evaluate speed-up optimization in the context of multiple sub-tasks, and do not extend their analysis to evaluate the relationship between task fitness and drafter, verifier divergence.

## C   ADDITIONAL EXPERIMENTS

**Drafter ablation.** We highlight, in Figure 10 that MGSM speed-up disparities persist across different drafter models. If we fix the verifier, Qwen2.5-3B, and ablate over drafter models we observe that slow languages are consistently slow, and fast languages are consistently fast. Notably Japanese is the slowest language across all tested drafters, and English is the fastest language in 50% of cases. This ablation further speaks to the consistency of multilingual speed-up disparities.

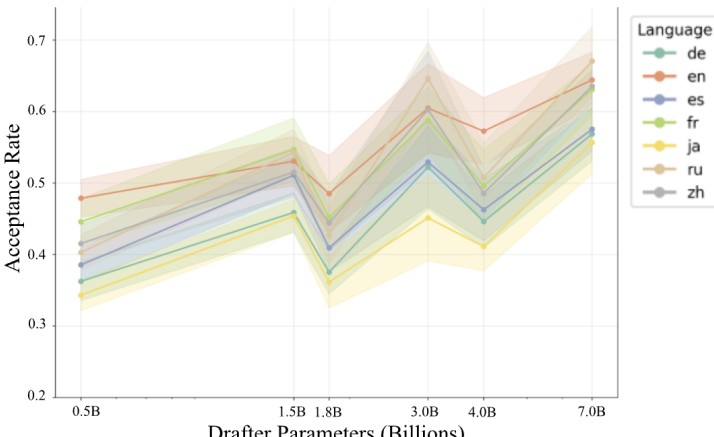

Figure 10: MGSM data, Qwen2.5 verifier (3B), various drafters: [Qwen2.5-(0.5B, 1.5B, 3B-base), and Qwen1.5-(4B, 7B)]. Acceptance scaling properties with drafter parameters. Speed-up hierarchy shows consistency across drafter models regardless of scale.

**Verifier ablation.** In Table 2 we ablate over verifier scales. We observe increases in initial unfairness as verifier size increases. We argue that this is because stronger verifiers (lower $r_p$) make the disparities in drafter fitness $r_q$ have more influence over speed-up unfairness, given Theorem 2. In applications where $P$ is larger, practitioners should be more cautious of unfairness issues.

**Verifier misalignment durign finetuning.** In Figure 11 we see the misalignment levels ($r_{p_T}$ for each task $T$), for the verifier (Qwen2.5-32B-Instruct), during s-CDF finetuning. We highlight that

Table 2: s-CDF scaling experiments for Qwen2.5 verifier from 1.5B to 32B, with Qwen2.5-0.5B drafter.

| Model | $\downarrow \mathcal{U}$ (Starting Unfairness) | $\downarrow \%\Delta\mathcal{U}$ (Unfairness Reduction) |
|---|---|---|
| P=Qwen2.5-1.5B | 0.004 | $-\mathbf{76.7\%} \pm 0.93$ |
| P=Qwen2.5-7B | 0.011 | $-72.9\% \pm 2.28$ |
| P=Qwen2.5-14B | 0.013 | $-56.4\% \pm 1.21$ |
| P=Qwen2.5-32B | 0.015 | $-69.8\% \pm 2.45$ |

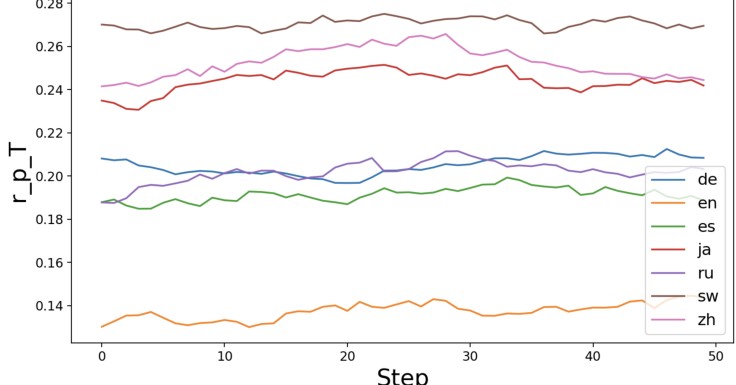

Figure 11: Verifier task-misalignment during s-CDF, showing the $P$ is frozen during training, and does not degrade in quality.

verifier performance is unaffected and thus (given Theorem A.5) generation quality remains constant during fairness mitigation.

## D    EXTENDED DISCUSSION AND REPRODUCIBILITY DETAILS

### D.1    S-CDF: TRAINING PROCEDURE

We implement s-CDF as a light-touch finetuning loop over languages (tasks) that (i) uses *teacher-forced completions from the verifier* as supervision for the drafter, and (ii) *scales gradients per-language* by their excess misfit over the current best language $r^\star$ to minimize variance from the minimum.[10]

**Models and quantization.**    We load a student/drafter $q$ and teacher/verifier $p$ with 4-bit NF4 quantization (bitsandbytes) and bfloat16 compute; the drafter enables gradient checkpointing and disables `use_cache` to reduce memory during training, while the verifier enables `use_cache` for fast generation.

**Data pipeline.**    We read a JSON of records with a `lang` field and question text; we index records per-language and tokenize prompts to a fixed `MAX_PROMPT_TOK`. Labels are retained on prompt tokens (non-padding) to compute a prompt-TV proxy later.

**Teacher completions and student loss.**    Given a mini-batch of prompts $X$, the teacher $p$ generates up to `MAX_GEN_TOKENS` tokens (sampling) with cache enabled; we splice out $S$ (new tokens) and form `seq_all = [X; S]` with a full-attention mask. The student $q$ is run on `seq_all` and trained with cross-entropy on the $S$ segment only (no prompt loss). This aligns $q$ to $p$'s next-token distribution on continuation tokens.

---

[10]Intuition: approximate the projected gradient of $\sum_T (D_T - D_{\min})^2$ where $D_T$ is task cross-entropy; this retains a certificate that increasing fitness raises a lower bound on $\alpha_T$ and therefore $S_T$.

**Per-language misfit and gradient shaping.** For each language $\ell$ sampled this step we compute $r_\ell = \mathrm{CE}(q\|p)$ (mean over the mini-batches of that language). We identify the current best language $\ell^\star = \arg\min_\ell r_\ell$ with value $r^\star$ and scale each $r_\ell$'s gradient by $(r_\ell - r^\star)/(\texttt{GRAD\_ACCUM} \cdot |\{\ell\}|)$ before updating $q$ (gradient clipping and accumulation supported). This implements the *shift-by-minimum* s-CDF update.

**Acceptance and prompt-TV logging.** At user-configured cadence we estimate acceptance rate with a *single-pass* proxy: draft $\gamma$ tokens with $q$ (no sampling), score the same positions with $p$, accept if $\min\{1, p/q\}$ exceeds a uniform random draw per token, and aggregate the *contiguous* accepted prefix length per row normalized by total drafted tokens. This tracks how alignment changes translate into realized acceptance. We also log a prompt-TV proxy for $p$ and $q$ ($1 - \Pr[\text{correct token}]$ on prompt positions ). Metrics are CSV-logged with timestamps.

**Optimizers and stability.** When quantized parameters are present we favor `AdamW8bit` (bitsand-bytes), otherwise standard `AdamW`. We clip gradients to `CLIP` before stepping every `GRAD_ACCUM` mini-batches.

**Default hyperparameters (reproducible starting point).** Unless stated, we used: `STEPS`=10,00; `SAMPLE_LANGS_PER_STEP`=5; `BATCH_PER_LANG`=64 prompts; `MINI_BATCH_SIZE`=8; `MAX_PROMPT_TOK`=512; `MAX_GEN_TOKENS`=64; `LR`=1e-4; `GRAD_ACCUM`=4; `CLIP`=1.0; acceptance draft width $\gamma = 5$ in the estimator.

## D.2 EXPERIMENTAL SETUP

**Hardware.** Training/ablation runs were conducted on single-GPU nodes (e.g., 1×A6000 48GB). The script constrains `max_memory` and uses 4-bit NF4 quantization for both $q$ and $p$ to fit comfortably.

**Software.** PyTorch (bf16 compute), HuggingFace `transformers`, bitsandbytes for quantization/optimizer; gradient checkpointing enabled on the drafter to control memory; `use_cache` toggled as described.

**Datasets and splits.** Following the paper, we evaluate multilingual math reasoning (MGSM, MCoT) and general instruction following (Dolly/Aya subset), reporting per-language acceptance/speed-ups and task metrics. See results sections and Appendix C for extended evidence of persistent disparities across drafters and datasets.

## D.3 MAIN RESULTS

**Multilingual speed-up disparities persist without mitigation.** Consistent with Section 4, acceptance and accuracy vary significantly by language; low-accuracy languages exhibit the lowest acceptance and thus the smallest speed-ups. We observe a persistent hierarchy across drafter scales (Qwen2.5 0.5–14B) with Japanese repeatedly among the slowest and English always the fastest, and the ordering is stable when swapping datasets (MGSM (Small) $\rightarrow$ MGSM (MCoT)).

**Throughput vs. quality.** Given that s-CDF never updates the verifier, the target distribution remains unchanged; by construction the drafter becomes a better proposal distribution w.r.t. $p$, and acceptance increases without altering $p$'s vanilla behavior. Quality metrics under vanilla decoding with $p$ remain stable; drafter-only generations improve on continuations seen during training due to student-on-teacher learning, but we do not use drafter-only decoding at inference time.

## D.4 REPRODUCIBILITY CHECKLIST

- **Data:** provide a JSON with fields `lang`, `question`; ensure each selected language has at least one record or it will be dropped.
- **Batching:** choose `SAMPLE_LANGS_PER_STEP`, `BATCH_PER_LANG`, `MINI_BATCH_SIZE` to fit memory; gradient-accumulate with `GRAD_ACCUM`.
- **Tokenization limits:** set `MAX_PROMPT_TOK`, `MAX_GEN_TOKENS`; pad-token is set to EOS if missing.

- **Optimizer:** use AdamW8bit when quantized, else AdamW; clip to `CLIP`.
- **Logging:** CSV columns include timestamp, step, `star_lang`, `lang`, $r_\ell$, acceptance, prompt-TV for $q$ and $p$.
- **Acceptance estimator:** keep default $\gamma=5$ (configurable); periodicity via `EVAL_EVERY`.

## D.5 LIMITATIONS AND PRACTICAL NOTES

**Estimator bias.** The logged acceptance proxy uses greedy drafts and per-token $\min\{1, p/q\}$ tests; it underestimates streak acceptance when deployment uses different temperatures or sampling filters. Use it for trend-tracking, not absolute certification.

**Data coverage.** When languages have very few prompts, $r^\star$ can be noisy; we recommend deploying in scenarios with large datasets. **Compute/latency.** Realized throughput depends on packing/verifier parallelism; speed-up equations assume negligible overhead when verifying $\gamma$-drafted tokens in one pass. Validate on your specific hardware.

However, note that s-CDF is quite simple to implement (by design), works with 4-bit quantized $q/p$, and *directly* targets the acceptance gap that governs speculative speed-ups.