# OpenReview forum: "The Disparate Impacts of Speculative Decoding"
_ICLR.cc/2026/Conference — Submitted to ICLR 2026_

### Official Review · Reviewer_zdoT · 2025-10-31

**Soundness:** 3
**Presentation:** 2
**Contribution:** 3
**Rating:** 6
**Confidence:** 2

**Summary:**

This paper investigates fairness issues in speculative decoding, where the inference speed-up achieved varies substantially across populations such as languages. Underrepresented or low-resource settings tend to experience slower decoding, revealing a form of computational disparity. The authors formalize this inequality using a divergence-based metric and introduce a mitigation method called stochastic corrective drafter finetuning (s-CDF). s-CDF is a straightforward approach that minimizes discrepancies. Empirical results demonstrate that the method effectively reduces these disparities, leading to fairer and more balanced acceleration across tasks.

**Strengths:**

* Novel framing: Reinterprets an efficiency-focused technique through a fairness perspective, exposing a blind spot in how LLM serving practices can create uneven user experiences.
* Theoretical grounding: Offers a formal connection between divergence, acceptance rate, and speed-up, framing fairness within speculative decoding in a principled way.
* Practical mitigation strategies: Demonstrates multiple, progressively stronger methods to reduce disparities, from simple adjustments like temperature tuning or data balancing to the more targeted optimization via s-CDF.

**Weaknesses:**

* The analysis is tightly bound to the Speculative Decoding, particularly with the drafter–verifier setup, making it unclear how well the ideas extend to other efficiency methods beyond speculative decoding.
* Some findings feel quite obvious. For instance, that higher acceptance rates lead to faster decoding, or that language representation in pretraining influences task performance.
* The proposed fairness metric, based on cross-entropy variance, may not directly correspond to user-perceived latency or practical deployment fairness.

**Questions:**

* How does speed-up unfairness manifest in real-world latency, cost differences, or overall user experience?
* Could this framework be extended to other acceleration methods, such as early exiting or mixture-of-experts inference?

---

> ### Author Response · Authors · 2025-11-20
> **Rebuttal To Reviewer zdoT**
>
> We thank the reviewer for the time spent diving into our work. We would like to offer some clarifications and additional experiments as per the questions and concerns. In light of the raised questions we have included additional experiments, and ablative baselines that serve to further support some of our decisions throughout the paper, and generally strengthen our work.
>
> **Weakness 1** *"The analysis is tightly bound to the Speculative Decoding, particularly with the drafter–verifier setup, making it unclear how well the ideas extend to other efficiency methods beyond speculative decoding."*
>
> This is an interesting point. However, we find that this exploration is more suitable for future work, given that our work directly addresses speculative-decoding, and the space of all efficiency methods is unfeasibly large for a single paper.
>
> **Weakness 2**  *"Some findings feel quite obvious. For instance, that higher acceptance rates lead to faster decoding, or that language representation in pretraining influences task performance."*
>
> We thank the reviewer for this feedback. These are not core findings of our work but rather instrumental points that we establish for further inference. Our work tries to avoid alienating readers by making these points, that may feel obvious to some, clear to all. The core findings of this paper center around the emperical and theoretic connections established between the representation of different tasks within a pretraining set, and the resulting speed-up of these tasks. We therefore find that disparities in pretraining representation motivate disparities in speculative speed-up, in ways that are non-trivial to predict or explain. Our work tries to relay to the community, for the first time, (i) the significant pitfalls of naively applying speculative-decoding, (ii) why certain social and culture groups may be affected, and (iii) derives a principled mechanism for supporting these affected sub-groups.
>
> **Weakness 3** *"The proposed fairness metric, based on cross-entropy variance, may not directly correspond to user-perceived latency or practical deployment fairness."*
>
> This a great point that we are happy to expand on. The cross entropy variance was choosen (i) because it provides a highly optimizable, precedented surrogate for mitgation, and (ii) connects theoretically to user-percieved latency ($S_T$) via:
>
> $$
>   S_T \geq \frac{f_\gamma(1-\sqrt{0.5\cdot\text{Cross-Entropy}_T})}{1+c\gamma},
> $$
>
> as derived in Theorem 1. The decision to use cross-entropy here is further supported by previous work in speculative acceleration (see DistillSpec Zhou et al. 2024) showing more desirable user-latency improvement. Finally, to address this concern further, we have added new ablations showing that cross-entropy has a stronger influence over user-percieved latency, and unfairness, than other available metrics, (see Tables 1, and 2).
>
> **Q1** *"How does speed-up unfairness manifest in real-world latency, cost differences, or overall user experience?"*
>
> This question addresses precisely why we find this work to be so impactful. Namely, we find that low representation, minority groups face systematically lower speed-ups during deployment. This means that applications that require low inference latency may only be suitable for majority groups. Additionally, minority group users of AI may suffer from a worse user experience, plaguged with higher inference times, and poor latency. This is an underdiscussed, and damaging problem that our work aims to illuminate and address.
>
> **Q2** *"Could this framework be extended to other acceleration methods, such as early exiting or mixture-of-experts inference?"*
>
> This is a very interesting question. Firstly, our framework is indeed extensible to any acceleration method whose speed-up scales with the divergence between two distributions $D_T$. This means in cases where an early exit model $Q$ is used to predict the final output $P$, we can hypothesize that the training representation used for $Q$, and $P$ will exhibit similar unfairness patterns as what we have observed in our work. Further, our mitigation tecnique should also be extensible to such cases. In general, these are meaningful directions for future research that our work helps to setup.
>
> --
> We thank the reviewer for their insightful comments. We hope that the additional details and results provided during the rebuttal have clarified all outstanding concerns, and we remain happy to address any further questions. We would also be sincerely grateful if you would consider updating the level of support for our paper accordingly.

---

> > ### Author Response · Authors · 2025-11-25
> >
> > Dear Reviewer,
> >
> > We thank you again for your review and engagement with our paper. As the discussion period is underway and ends in about a week, we look forward to further engaging with you to make sure all of your concerns are addressed.
> >
> > Looking forward to a fruitful discussion period!
> >
> > Sincerely,
> >
> > The Authors

---

### Official Review · Reviewer_y91N · 2025-11-01

**Soundness:** 2
**Presentation:** 3
**Contribution:** 2
**Rating:** 4
**Confidence:** 4

**Summary:**

This paper points out that the speedup performance of speculative decoding is uneven across different tasks and languages, with slower speedups on tasks with low fit and insufficient representativeness (such as low-resource languages). The authors quantify this unfairness, revealing its correlation with the fit of the draft model, and propose a randomized correction fine-tuning strategy that improves the fairness metric by an average of 12%.

**Strengths:**

- The paper elucidates this from three points: (i) why speed-up is monotone in acceptance, (ii) how speed-up is consequently induced by drafter-verifier fitness, and (iii) how cross-entropy misfit provides an optimizable surrogate for speed-up unfairness with provable implications for acceleration. The structure is very clear.

- The paper's definition of unfairness is very clear and reasonable.

**Weaknesses:**

- The main experiment only compared English and Japanese, which is too limited in terms of language compatibility. It is recommended to conduct experiments using 3-4 languages.

- In the experiment, the small and large models were from the same series, such as Qwen2.5-0.5B and Qwen2.5-3B. There is a lack of comparison for cases where the small and large models are not from the same series.

**Questions:**

- This paper primarily focuses on inequalities in efficiency/time, but in practical use, users are more concerned with the effectiveness. Does inequality exist in terms of effectiveness?

- If the verifier is larger, such as Qwen3-32B, will the results be different, considering that the verifier's parameter size is likely larger in real-world applications?

---

> ### Author Response · Authors · 2025-11-20
> **Rebuttal To Reviewer  y91N**
>
> We thank the reviewer for the time spent diving into our work. We would like to offer some clarifications and additional experiments as per the questions and concerns. In light of the raised questions we have included additional experiments, and ablative baselines that serve to further support some of our decisions throughout the paper, and generally strengthen our work.
>
> **Weakness 1** *"The main experiment only compared English and Japanese, which is too limited in terms of language compatibility. It is recommended to conduct experiments using 3-4 languages."*
>
> This point is definetly worthy of elaboration. The English, Japanese experiment serves as a motivating example to show that bi-lingual mitigation is possible. We later build on this example when deploying our s-CDF algorithm. In this case we extend to five langauges to convey our message. Additionally, we have run further experiments to align with this request, expanding our analyis to seven languages (see Tables, 1 and 2).
>
> **Weakness 2** *"In the experiment, the small and large models were from the same series, such as Qwen2.5-0.5B and Qwen2.5-3B. There is a lack of comparison for cases where the small and large models are not from the same series."*
>
> We thank the reviewer for this note. We designed our experiments to reflect as much as possible realistic speculative-decoding unfairness issues. Given that speculative-decoding requires both models to be from the same series (for identical tokenizers), we have restricted our analysis to this case. While we would like to investigate models from different series, vanilla speculative-decoding makes this impossible.
>
> **Q1** *This paper primarily focuses on inequalities in efficiency/time, but in practical use, users are more concerned with the effectiveness. Does inequality exist in terms of effectiveness?*
>
> We appreciate this important question. Interstingly speculative-decoding offers theoretic gaurantees that 'effectiveness' does not get degraded during deployment. We have included a figure to highlight this in Appendix C (Figure 11; showing that verifier loss $r_p$ stays constant during mitigation), which paired with the proof in Appendix A.5 gaurantees consistent effectiveness.
>
> **Q2** *If the verifier is larger, such as Qwen3-32B, will the results be different, considering that the verifier's parameter size is likely larger in real-world applications?*
>
> This is an intersting point. Theorem 2 shows that drafter fitness ($1-r_q$) becomes a strong predictor of speed-up $\alpha_T$, the more fit the verifier $p$ is. Specifically
> $$
>   (1-r_q) = \alpha_T \pm r_p,
> $$
>
> for verifier loss $r_p$. To relate this to the question, when the verifier $p$ becomes larger, our analysis allows us to infer that (given that $r_p$ becomes lower) the drafter alignment $r_q$ plays a more critical role in dictating speed-up. In other words, if the verifier is larger, then issues of drafter bias (varying $r_q$ across tasks) will impact speed-up unfairness more. To answer this question emperically, we have incorporated results from scaling up the verifier to 32B parameters (see Table 3). We see that when scaling up the verifier, fairness issues become more pronounced (with a $\sim3\times$ increase from 1.5B to 32B in unfairness). Importantly, we also see that the proposed mitigation strategy (s-CDF) generalizes to the cases, resulting in consistent unfairness reduction.
>
> Table 3: s-CDF scaling experiments for Qwen2.5 verifier from 1.5B to 32B, with Qwen2.5-0.5B drafter.
>
> | Model                          | $\downarrow\mathcal{U}$ (Starting Unfairness) | $\downarrow\%\Delta \mathcal{U}$ (Unfairness Reduction) |
> |----------------------------------|----------------------------------------------------------|------------------------------------------|
> | P=Qwen2.5-1.5B                | 0.004                                             | $\mathbf{-76.7\%}\pm 0.93$           |
> | P=Qwen2.5-7B       | 0.011                                                 | $-62.9\%\pm 2.28$                    |
> | P=Qwen2.5-14B                      | 0.013                                        | $-56.4\%\pm 1.21$                    |
> | P=Qwen2.5-32B              | 0.015                                                 | $-49.8\%\pm 2.45$
>
>
> ---
> We thank the reviewer for their insightful comments. We hope that the additional details and results provided during the rebuttal have clarified all outstanding concerns, and we remain happy to address any further questions. We would also be sincerely grateful if you would consider updating the level of support for our paper accordingly.

---

> > ### Comment · Reviewer_y91N · 2025-11-28
> >
> > Thank you for your responses. After reading them, I realized that my Weakness 2 was not well-founded, and I apologize for that. I am satisfied with your replies and will raise my score to 6 as soon as OpenReview allows editing. However, I still believe that most of the main experiments should involve more than two languages, and I encourage the authors to consider adding such experiments in a future version of the paper.

---

> ### Author Response · Authors · 2025-11-25
> **Follow-up and clarification**
>
> Dear Reviewer,
>
> as the discussion period is nearing its end, we wanted to ask if there are any follow-up points we can clarify.
>
> We believe we have responded to all questions and concerns raised, in addition to taking a few days to run the additional experiments necessary to demonstrate our points, all of which are incorporated in the revised version of the paper.
> \
> If there are no further points of clarification regarding the manuscript, we kindly ask that reviewer y91N consider increasing their score to reflect the improvements and clarifications we have provided. We are happy to continue to engage in discussion and answer any questions!

---

### Official Review · Reviewer_4XQp · 2025-11-01

**Soundness:** 3
**Presentation:** 2
**Contribution:** 2
**Rating:** 4
**Confidence:** 4

**Summary:**

This paper aims to explore the different amount of inference speed up realized by the speculative decoding on different tasks/languages. Towards this, the paper first empirically demonstrates the existence of such a disparate latency reduction across languages and identify the correlation between the poor latency reduction and final model accuracy. The paper then develops a theoretical framework to study this phenomenon (which the paper refers to as unfairness in speculative decoding) and connects the acceptance rate for draft tokens to per-task divergence between the drafter & verifier distributions as well as the task fitness of the drafter model. The paper propose a solution by minimizing the variance in the per-task divergences between the drafter & verifier model.

**Strengths:**

- The paper focuses on a rather unexplored yet important topic of disparate impacts of speculative decoding across different tasks/languages.
- The paper successfully demonstrates the existence of poorer latency reduction for rarer languages.
- The aims to develop a rigorous theoretical framework to understand and tackle the unfairness in latency reductions via speculative decoding across different tasks/languages.

**Weaknesses:**

- The novelty and significance of theoretical contributions in the paper appears to be somewhat limited. Theorem 1 looks like a straightforward corollary of the existing results about speculative decoding in the literature (e.g., see Leviathan et al.). Also, it's not clear why the authors presented the lower bound on $S_T$ in the form of $K_T$ or $D_T$. What is the downside of keeping the lower bound in the form of total variation distance, i.e., the first inequality in Eq. (5)? Similarly, Theorem 2 is just a triangle inequality. Could the authors clarify where the assumption $r_p < r_q$ is used in the proof of Theorem 2.
- The paper is missing important baselines. The paper has identified that rarer languages benefit less from speculative decoding. This would naturally suggest that one should train drafter model in such a manner that its performance is relatively more uniform across different tasks/languages. Admittedly, the authors have considered data balancing during the drafter training in Section 8.3. However, the exploration is not comprehensive enough. There is a vast literature on long-tail learning which proposed custom loss functions to ensure good model performance for both rare vs. popular subpopulations (tasks/languages). The authors should consider expanding their empirical section by studying the effect of some of the techniques from this literature. Furthermore, how would techniques like DistillSpec affect the unfairness is not studies and compared with the proposed s-CDF method.

**Questions:**

See weaknesses section above. The reviewer has a few additional questions:

- Lines 266-269 discuss dropping a terms from $\nabla\_{\theta}\mathcal{U}$. Did you empirical verify the utility of dropping this term in your submission?
- In Line 431, the authors mention ``We see on average, a 20\% reduction...a 12\% decrease in unfairness $\mathcal{U}$.`` Are these results documented (in the form of figures or tables) in the submission?
- Potential typo in Line 128 - `... **tb** section 6 and 8`.

---

> ### Author Response · Authors · 2025-11-20
> **Rebuttal To Reviewer 4XQp (Part 1/4)**
>
> We thank the reviewer for the time spent diving into our work. In light of the raised questions we have included additional experiments and ablative baselines that serve to further support some of our decisions throughout the paper, and to strengthen our work (changes made in blue). Prior to responding to the questions in detail, we want to emphasize that our work does not focus primarely on mitigation techniques, nor is it designed to perform an in-depth assessment of differing mitigation strategies for speculative decoding unfairness. Instead, our work (1) *sheds light* on an under-discussed issue of speculative-decoding unfairness, (2) derives a theoretic analysis to *quantify and explain* the emergence of unfairness, and (3) offers an evaluation framework, and *principled mitigation strategy* that we believe will serve as the basis for *future mitigation work* from this community.
>
> Table 1: Finetuning ablations (For Qwen2.5-0.5B -> 1.5B) showing change in average acceptance, unfairness, and effect on fastest language.
>
> | Method                           | $\uparrow\%\Delta \mathrm{Mean}(\alpha_{T})$ (Avg. Acceptance Increase) | $\uparrow\%\Delta \alpha_{\max}$ (Fastest Lang. Increase) | $\downarrow\%\Delta \mathcal{U}$ (Unfairness Decrease) |
> |----------------------------------|----------------------------------------------------------|------------------------------------------|--------------------------------------|
> | **s-CDF (ours)**                | $+5.3\%$                                                 | $+3.2\%$                                 | $\mathbf{-76.7\%}\pm 0.93$           |
> | s-CDF (w/ Degrading Term)       | $-0.1\%$                                                 | $-0.7\%$                                 | $-19.4\%\pm 1.62$                    |
> | DistillSpec                      | $\mathbf{+5.8\%}$                                        | $\mathbf{+4.1\%}$                         | $-64.0\%\pm 2.15$                    |
> | s-CDF (w/ TV Loss)              | $+4.4\%$                                                 | $+1.8\%$                                 | $-30.3\%\pm 1.36$                    |
>
> Table 2: Finetuning ablations (For Qwen2.5-0.5B -> 3B) showing change in average acceptance, unfairness, and effect on fastest language.
>
> | Method                           | $\uparrow\%\Delta \mathrm{Mean}(\alpha_{T})$ (Avg. Acceptance Increase) | $\uparrow\%\Delta \alpha_{\max}$ (Fastest Lang. Increase) | $\downarrow\%\Delta \mathcal{U}$ (Unfairness Decrease) |
> |----------------------------------|----------------------------------------------------------|------------------------------------------|--------------------------------------|
> | **s-CDF (ours)**                | $+5.1\%$                                                 | $+2.91\%$                                 | $\mathbf{-78.1\%}\pm 0.63$           |
> | s-CDF (w/ Degrading Term)       | $-0.3\%$                                                 | $-0.8\%$                                 | $-12.9\%\pm 1.28$                    |
> | DistillSpec                      | $\mathbf{+5.7\%}$                                        | $\mathbf{+3.3\%}$                         | $-62.4\%\pm 2.21$                    |
> | s-CDF (w/ TV Loss)              | $+4.2\%$                                                 | $+1.5\%$                                 | $-29.8\%\pm 1.45$                    |
>
> Our tested methods include the s-CDF algorithm from the paper (Algorithm 1), the s-CDF algorithm with the 'Degrading Term' included (see section 7), DistillSpec (uniform weights across tasks, [Zhou et al. 2024]) and finally s-CDF where cross-entropy is replaced with TV loss.
>
> (Part 1/4, Continued Below)

---

> ### Author Response · Authors · 2025-11-20
> **Rebuttal To Reviewer 4XQp (Part 2/4)**
>
> These new results exhibit behavior consistent with the discussion in our submission. Firstly, we see that including the degrading term ($-(D_T - D_{\min})\nabla_\theta D_T$) causes a reduction in the speed-up of the fastest language (English; $-0.7\%$ for 1.5B $P$), as well as leads to less unfairness convergence than s-CDF (likely due to instability introduced by unlearning). Secondly, we see that uniform DistillSpec results in large mean and fastest-lang increases, as expected, yet produces *sub-optimal unfairness-mitigation*, exhibiting a $-12\%$ lower mitigation magnitude than s-CDF. As argued in the paper, we interpret this to be because minority and majority langauges are equally weighted in the objective, with no notion of disparities incldued. We have included these results, and associated discussion, in the revised submission (see Section 8.4). We will continue to expand on these results as they pertain to the subsequent questions.
>
> However, we want to emphasize that our work does not focus primarily on mitigation techniques, nor is designed to perform in-depth assessment of differing mitigation strategies for speculative decoding unfairness. Shortly, our work focuses on (1) *shedding light* on an under-discussed issue of speculative-decoding unfairness, (2) deriving a theoretic analysis to *quantify and explain* the emergence of unfairness, and (3) offering an evaluation framework, and *principled mitigation strategy* that we believe will serve as the basis for *future mitigation work* from this community. We will continue to empahsize these points where relavent.
>
> **Weakness 1a** *The novelty and significance of theoretical contributions in the paper appears to be somewhat limited. Theorem 1 looks like a straightforward corollary of the existing results about speculative decoding in the literature (e.g., see Leviathan et al.).*
>
> We are grateful for this input. We argue that this is perhaps an issue of presentation. We do not claim Theorem 1 to be a core contribution of our work, but rather an inference that was included to set-up later discussion. Our work tried to avoid alienating readers unfamiliar with speculative-decoding literature like (Leviathan et al. 2023), to keep the paper **self-contained**. We have also attempted to emphasize in our submission that Theorem 1 is based on prior work, and have updated the language to emphasize this further. Generally, our **theoretic contributions** center around (i) *formalizing fairness issues*, (ii) providing grounds for *principled mitigation*, and (iii) offering *theoretic explanations* for the observed empirical trends. These are points we will continue to emphasize as they relate to the later queries.
>
> **Weakness 1b** *"...Theorem 2 is just a triangle inequality. Could the authors clarify where the assumption $r_p \leq r_q$ is used in the proof of Theorem 2."*
>
> We once again appreciate the opportunity to expand on this. Theorem 2 is in fact critical for understanding the motivating factors behind speed-up unfairness, and while it may appear theoretically elementary, in the context of speculative unfairness we find it powerful for illuminating the connection between task representation and speed-up unfairness. Namely, we yield the insight that $1-r_q = \alpha_T \pm r_p$ for task acceptance $\alpha_T$, and model losses $r_p, r_q$. Specifically for the speculative-decoding use case, $r_p$ tends to be low (given that $p$ is a large, fit model) thus drafter fitness $1-r_q$, becomes a strong determinant of task speed-up ($1-r_q \approx \alpha_T$ with likely small error $r_p$). This offers the insight that in practice, drafter bias can significantly motivate speed-up unfairness. We argue that this provides the community with foresight into when speed-up unfairness issues are likely to emerge, and why they occur in practice.
>
> **Weakness 1c** *"... "What is the downside of keeping the lower bound in the form of total variation distance, i.e., the first inequality in Eq. (5)?"*
>
> This is another subtle point that we are happy to have the chance to expand on. We find that the TV term is in fact not ideal for optimization, (as reported in DistillSpec; Zhou et al. 2024). Using CE divergence further yields an optimization strategy more compatible with exisitng distillaiton paradigms. Our own ablative baselines reveal in practice, (Tables 1, and 2 above), that the TV loss serves to produce lower convergence in acceptance and thus unfairness, relative to the CE Divergence.
>
> (Part 2/4, Continued Below)

---

> ### Author Response · Authors · 2025-11-20
> **Rebuttal To Reviewer 4XQp (Part 3/4)**
>
> **Weakness 2a** *"The paper is missing important baselines. The paper has identified that rarer languages benefit less from speculative decoding. This would naturally suggest that one should train drafter model in such a manner that its performance is relatively more uniform across different tasks/languages."*
>
> This is an excellent point that we have taken into account to further improve the paper. We have included an ablative analysis of differing mitigation methods to justify some of our discussed decisions emperically (Tables 1 and 2 above), aligning with our discussion, theory, and intuition provided in the submission. We will continue to expand on these results throughout the response.
>
> However, we would also like to re-emphasize that our work is not designed to be a nuanced emperical analysis of differing mitagation methods, but rather point out a potentially damaging, and under-discussed trend of speed-up unfairness. We derive a framework for quantifying this unfairness that leads to a prinicpled mitagation method, (s-CDF), and while we are unable to find more suitable approaches in our analysis, we do not claim, nor expect, s-CDF to be the optimal approach. We leave this question of optimal mitigation open for future research to build on our framework.
>
> **Weakness 2b** *"There is a vast literature on long-tail learning which proposed custom loss functions to ensure good model performance for both rare vs. popular subpopulations (tasks/languages). The authors should consider expanding their empirical section by studying the effect of some of the techniques from this literature."*
>
> We thank the reviewer for this relavent concern. Shortly, we aimed to approach the issue of mitigation in our work in as **principled** a manner as possible. We find that the field of long-tail learning is somewhat distinct from this, frequently using heuristic-guided class-balancing protocols [Zhang et al. 2023]. In fact, applying our s-CDF mitigation stochastically bears an emergent resemblence to such long-tail methods. For instance, in Figure 8. we show that determining that majority (fastest) task $T_{\min}$, in a mini-batch wise manner results in a natural 'downsampling/upsampling' effect, where English, for example, is downsampled to only $\sim 7\%$ of batches (recieving no gradient update in the $\sim 93\%$ of other cases). However, this downsampling has occured in a **heurestic-free** manner, as this probability is emergent from applying the principled, true unfairness minimizer (Equation 9), stochastically.
>
> We would further like to emphasize that our work does not attempt to improve upon the methods within the vast long-tail learning field, nor does it present itself as an optimal long-tail mitigation approach. As the reviewer has identified, there is already extensive literature in this area, and the novelty of such an analysis is limited. Rather, our work is intended to shed light on this very *specific*, underdiscussed and impactful issue of speed-up unfairness, present scalable, principled mitigation, as well as seed the subseqeuent development of further strategies via our introduced formalizms.
>
> **Weakness 2c** *"...how would techniques like DistillSpec affect the unfairness is not studies and compared with the proposed s-CDF method"*
>
> This is an important point to establish emperically. Our ablative baselines in Tables 1, and 2 showcase the differing influence of DistillSpec and s-CDF based methods. We see that DistillSpec more directly maximizes average acceptance, and majority acceptance as expected, but at the cost of unfairness reduction (e.g., Table 1 shows $-12\%$ less unfairness reduction with DistillSpec than s-CDF). We find that s-CDF remains a more principled and emperically effective approach for targetting speed-up unfairness directly.
>
> **Q1** *"Lines 266-269 ... Did you empirical verify the utility of dropping this term in your submission?"*
>
> Similarly, in Tables 1 and 2 we see that the emperical results match our expectations. Including the degrading term ('s-CDF w\ Degrading Term' above) results in a slight reduction in the majority speed-up (Table 1; $-0.7\%$ in English acceptance), and negligable change in average speed (Table 1; $-0.1\%$ in average acceptance). As anticipated the negative gradient term $-(D_T - D_{\min})\nabla_\theta D_T$ consistently introduces additional instability (via unlearning), and directly degrades drafter quality.
>
> (Part 3/4, Continued Below)

---

> ### Author Response · Authors · 2025-11-20
> **Rebuttal To Reviewer 4XQp (Part 4/4)**
>
> **Q2** *"In Line 431, the authors mention ... we see on average, a 20\% reduction...a 12\% decrease in unfairness $\mathcal{U}$. Are these results documented (in the form of figures or tables) in the submission?"*
>
> Our more in depth emperical investigations detail fairness mitigation more clearly in the form of Tables 1, 2 (Section 8.4). Given the greater number of distillation tokens, the new results feature unfairness reductions up to $\sim 74\%$, and the paper has been updated to reflect this. These are calculated by compute the unfairness $\%$ change in $\cal U$ during mitigation.
>
> ---
> We thank the reviewer for their insightful comments. We hope that the additional details and results provided during the rebuttal have clarified all outstanding concerns, and we remain happy to address any further questions. We would also be sincerely grateful if you would consider updating the level of support for our paper accordingly.
>
> **References**
>
> [Zhou et al. 2024]: Yongchao Zhou et al. "DistillSpec: Improving Speculative Decoding via Knowledge Distillation", ICLR 2024
>
> [Zhang et al. 2023]: Yifan Zhang et al. "Deep Long-Tailed Learning: A Survey", IEEE 2023

---

> ### Author Response · Authors · 2025-11-25
>
> Dear Reviewer,
>
> as the discussion period is nearing its end, we wanted to ask if there are any follow-up points we can clarify.
>
> We believe we have responded to all questions and concerns raised, in addition to taking a few days to run the additional experiments necessary to demonstrate our points, all of which are incorporated in the revised version of the paper.
> \
> If there are no further points of clarification regarding the manuscript, we kindly ask that reviewer 4XQp consider increasing their score to reflect the improvements and clarifications we have provided. We are happy to continue to engage in discussion and answer any questions!

---

### Official Review · Reviewer_XiyQ · 2025-11-04

**Soundness:** 2
**Presentation:** 4
**Contribution:** 3
**Rating:** 4
**Confidence:** 4

**Summary:**

Speculative decoding speeds up sampling of a large (verifier) LLM by having a smaller (drafter) model generate an output sequence, which is then verified by the verifier model. Key to the latency reduction is that the verification step over sequence prefixes can be done in parallel. This paper makes an observation that the speed-up gained is not uniform across all tasks. For instance, the speed-up achieved on Japanese language tasks is less compared to the speed-up on English. The paper makes a connection between the proportion of data from a task (e.g., a language) and the speed-up achieved from speculative decoding. Briefly and roughly, rare tasks (as measured by their presence in the training set) tend to yield less speed-up. The paper also proposes a gradient-based mitigation strategy (Sec 7).

**Strengths:**

While speculative decoding has gathered much attention as of late, to my knowledge, this work is one of the first that attempts to formally connect the occurrence probability of a task to the per-task speed-up from speculative decoding. The research direction is original. The paper is mathematically precisely written, and is easy to read overall. Sections are also organized appropriately. Connecting the task speed-up to the cross entropy between the two models on data of that task gives interesting insights (Theorem 1). Though the results only rely on basic inequalities, there is value from the precision in writing, and good explanation.

**Weaknesses:**

While the paper is very well written and the theoretical results are interesting, the paper misses the following important points.

1. The paper lacks baselines in experiments. I see only one baseline method which is “Data balancing” (L417). This is understandable since the paper tackles a new sub-area. However, I think a few “ablative” baselines would provide more insights.

2. The abstract and the formulation consider generic notion of tasks. However, experiments consider only one definition, which is that one task is one natural language. Investigating alternate notions of tasks would provide more insights.

More specific questions are given in Questions. Overall, I find the research direction and theoretical results promising.  It is just that the empirical results do not quite provide sufficient evidence.

**Questions:**

**Questions**:

**Q1**:  Are there other simple baseline methods that can be compared to? By baseline, I mean a method that attempts to provide more uniform speculative decoding speed-up to all tasks. Currently there is only one baseline (Data Balancing). Please correct me if I’m wrong.


**Q2**:  Are there other notions of tasks to show the generality of the theoretical results and the proposed unfairness mitigation approach? Currently, in experiments, one task is defined as a data subset that has text from one language. It would be interesting to consider tasks that have overlapping support. That is, a prefix $s \\in \\mathcal{V}^*$  (per L137, Sec 5) can belong to more than one task (with non-trivial probabilities).


**Q3**:  In the paragraph after Theorem 2 at line 209, we have

> drafter fitness $(1- r_q) \\uparrow \\implies \\alpha_T \\uparrow$

Could you please elaborate on this implication? I don’t quite see why $(1- r_q)$ increasing would imply $\\alpha_T$ increasing. I may be missing something trivial.

**Q4**: Is it possible to have a frequent task $T$ (i.e., a significant portion in the training set) that has a large $D_T$ (i.e., large discrepancy between $q$ and $p$ on the task $T$)? Could you please give an example? In that case, what are the effects of the proposed procedure on the speed-up on task $T$?


**Q5**: Related question. What happens to the speed-up gained on task $T$ if it is rare in the training set of $q$ (drafter), and frequent in the training set of $p$ (verifier)? And vice versa? This question is not about the proposed mitigation strategy. The setting is that $q$ and $p$ are given (where $q$ may not have been trained to align with $p$). Does the non-uniformity of tasks in the training of $q$ (or $p$) have more negative effects on the speed-up gain?


**Q6**:  I understand that the proposed mitigation strategy will trade-off speed-up gains across tasks. I don’t think this point is sufficiently elaborated. **This is an important point.** Take the example considered in the paper where there are two tasks: English (dominant) and Japanese (rare). After using the proposed procedure, what happens to the speed-up on the English task? Looking at the proposed gradient in Sec 7, L258, weighting across $m$ tasks is uniform $1/m$. For a dominant task whose base occurrence probability is above $1/m$, it will be weighted down. Is this correct?


**Q7**: Is it correct that the proposed fairness mitigation strategy does not change the overall quality (across all tasks)? This is because the procedure freezes $p$ (verifier), and only fine-tunes $q$, and speculative decoding guarantees that the sample generated will follow $p$, regardless of $q$. Is this correct? If so, it may be worth highlighting, say, in Sec 8 (experiments). It may be trivial to those working on speculative decoding but it is helpful for non-specialist readers.


I very much hope all the above questions can be addressed.


----------

**Minor issues** (easily fixable):

* Duplicate references to Leviathan et al., 2023.

* L215: preforms -> perform

---

> ### Author Response · Authors · 2025-11-20
> **Rebuttal To Reviewer XiyQ (Part 1/3)**
>
> We thank the reviewer for the time spent diving into our work. In light of the raised questions we have included additional experiments and ablative baselines that serve to further support some of our decisions throughout the paper, and to strengthen our work (changes made in blue). Prior to responding to the questions in detail, we want to emphasize that our work does not focus primarely on mitigation techniques, nor is it designed to perform an in-depth assessment of differing mitigation strategies for speculative decoding unfairness. Instead, our work (1) *sheds light* on an under-discussed issue of speculative-decoding unfairness, (2) derives a theoretic analysis to *quantify and explain* the emergence of unfairness, and (3) offers an evaluation framework, and *principled mitigation strategy* that we believe will serve as the basis for *future mitigation work* from this community.
>
> **Weakness 1/Q1**  *"The paper lacks baselines in experiments."*
>
> While we remark once more that our work does not aim to offer an in-depth assessment on mitigations (but is more focused on highlighting and explaining why speculatative decoding may exacerbate unfairness), we have included the ablative baselines below that align with our provided intuition and theoretical insights.
>
> Table 1: Finetuning ablations (For Qwen2.5-0.5B -> 1.5B) showing change in average acceptance, unfairness, and effect on fastest language.
>
> | Method                           | $\uparrow\%\Delta \mathrm{Mean}(\alpha_{T})$ (Avg. Acceptance Increase) | $\uparrow\%\Delta \alpha_{\max}$ (Fastest Lang. Increase) | $\downarrow\%\Delta \mathcal{U}$ (Unfairness Decrease) |
> |----------------------------------|----------------------------------------------------------|------------------------------------------|--------------------------------------|
> | **s-CDF (ours)**                | $+5.3\%$                                                 | $+3.2\%$                                 | $\mathbf{-76.7\%}\pm 0.93$           |
> | s-CDF (w/ Degrading Term)       | $-0.1\%$                                                 | $-0.7\%$                                 | $-19.4\%\pm 1.62$                    |
> | DistillSpec                      | $\mathbf{+5.8\%}$                                        | $\mathbf{+4.1\%}$                         | $-64.0\%\pm 2.15$                    |
> | s-CDF (w/ TV Loss)              | $+4.4\%$                                                 | $+1.8\%$                                 | $-30.3\%\pm 1.36$                    |
>
> Table 2: Finetuning ablations (For Qwen2.5-0.5B -> 3B) showing change in average acceptance, unfairness, and effect on fastest language.
>
> | Method                           | $\uparrow\%\Delta \mathrm{Mean}(\alpha_{T})$ (Avg. Acceptance Increase) | $\uparrow\%\Delta \alpha_{\max}$ (Fastest Lang. Increase) | $\downarrow\%\Delta \mathcal{U}$ (Unfairness Decrease) |
> |----------------------------------|----------------------------------------------------------|------------------------------------------|--------------------------------------|
> | **s-CDF (ours)**                | $+5.1\%$                                                 | $+2.91\%$                                 | $\mathbf{-78.1\%}\pm 0.63$           |
> | s-CDF (w/ Degrading Term)       | $-0.3\%$                                                 | $-0.8\%$                                 | $-12.9\%\pm 1.28$                    |
> | DistillSpec                      | $\mathbf{+5.7\%}$                                        | $\mathbf{+3.3\%}$                         | $-62.4\%\pm 2.21$                    |
> | s-CDF (w/ TV Loss)              | $+4.2\%$                                                 | $+1.5\%$                                 | $-29.8\%\pm 1.45$                    |
>
> Our tested methods include the s-CDF algorithm from the paper (Algorithm 1), the s-CDF algorithm with the 'Degrading Term' included (see section 7), DistillSpec (uniform weights across tasks [Zhou et al. 2024]) and finally s-CDF where cross-entropy is replaced with TV loss.
>
> (Part 1/3)

---

> ### Author Response · Authors · 2025-11-20
> **Rebuttal To Reviewer XiyQ (Part 2/3)**
>
> These new results (Tables 1 and 2 above) exhibit behavior consistent with the discussion in our submission. Firstly, we see that including the degrading term ($-(D_T - D_{\min})\nabla_\theta D_T$) causes a reduction in the speed-up of the fastest language (English; $-0.7\%$ for 1.5B $P$), as well as leads to less unfairness convergence than s-CDF (likely due to instability introduced by unlearning). Secondly, we see that uniform DistillSpec results in large mean and fastest-lang increases, as expected, yet produces *sub-optimal unfairness-mitigation*, exhibiting a $-12\%$ lower mitigation magnitude than s-CDF. As argued in the paper, we interpret this to be because minority and majority langauges are equally weighted in the objective, with no notion of disparities incldued. We have included these results, and associated discussion, in the revised submission (see Section 8.4). We will continue to expand on these results as they pertain to the subsequent questions.
>
> **Weakness 2/Q2** *"The abstract and the formulation consider generic notion of tasks. However, experiments consider only one definition..."*
>
> This a good point. Our framework is agnostic to the task at hand but we have focused our empirical analysis on the occurrence of speculative-decoding disparities with the most significant negative externalities. We argue that the multilingual setting is one where the *social and cultural implications* of speed-up unfairness are most destructive, and thus other forms of tasks, while suitable for future work, may be slightly orthogonal to the core message of our paper. We are happy to revise our writing in the abstract and introduction to emphasize this message if the reviewer suggests it.
>
> **Q2** *"...It would be interesting to consider tasks that have overlapping support. That is, a prefix  (per L137, Sec 5) can belong to more than one task (with non-trivial probabilities)."*
>
> Thank you for this question. We have added a proof in Appendix A.6, showing that for two tasks $T1, T2$ increasing the overlap between the tasks decreases the disparities $|D_{T1}-D_{T2}|$. Formally, let $p_\cap := 1 - TV(T1, T2)$, denote the overlap probability. We show that increasing $p_{\cap}$ decreases the disparity $|D_{T1}-D_{T2}|$ (i.e., $\frac{\partial |D_{T_1} - D_{T_2}|}{\partial p_{\cap}}<0$). This is intuitive given that if $p_{\cap}=1$ then the tasks are the same (i.e., $T_1 = T_2$), and so the disparities do not exist (i.e., $|D_{T_1} - D_{T_2}|=|D_{T_1} - D_{T_1}|=0$).
>
> **Q3** *"In the paragraph after Theorem 2 at line 209, ... could you please elaborate on this implication?"*
>
> Absolutely. This equation is one of our key theoretic and empirical findings. The proof is featured in Appendix A.3 and shows that the error of the approximation $1-r_q \approx \alpha_T$ is bounded by $r_p$, i.e.,
> $$
>     (1-r_q) - \alpha_T \leq r_p, \\
>     \text{or, } 1-r_q = \alpha_T \pm r_p.
> $$
> We further see this tendency empirically in Figure 2. Intuitively, this says the drafter fitness on some task $T$ closely controls the speed-up on task $T$ when verifier fitness $r_p$ is high. We offer this as an explanation for why tasks with high representation (and thus lower task loss $r_q$), are more likely to exhibit higher speed-up (high $\alpha_T$).
>
> **Q4** *"Is it possible to have a frequent task $T$ that has a large divergence)? ... In that case, what are the effects of the proposed procedure on the speed-up on task T?"*
>
> To clarify, this question asks is if a task $T$ with high representation can also be slow, and what the downstream effects are. To formalize, we would say that $r_{q_T}$ and $r_{p_T}$ (drafter, verifier loss levels on $T$) are relatively low (given that more data pushes task training loss lower). We can then theoretically bound task-acceptance $\alpha_T$, and say that the worst case $\alpha_T$ is defined by $\alpha_T \geq 1-\frac{1}{2}(r_{q_T}+r_{p_T})$ (see Appendix A.7). So, the higher representation $T$ is, the lower $(r_{q_T}+r_{p_T})$ is, and thus the higher the worst case $\alpha_T$ is. For example, we again refer to Figure 2 for the empirical tendency of $\alpha_T$ to correlate with $1-r_q$. In short, yes it is possible that $T$ exists as described, however, we can bound the worst case divergence, which will generally increase with the representation of $T$ in $q$ and $p$.
>
> (Part 2/3)

---

> ### Author Response · Authors · 2025-11-20
> **Rebuttal To Reviewer XiyQ (Part 3/3)**
>
> **Q5** *"Related question. What happens to the speed-up gained on task  if it is rare in the training set of  (drafter), and frequent in the training set of  (verifier)? And vice versa? ..."*
>
> Great question. We formalize this by talking about the distance $|r_q-r_p|$ of model fitness levels (as this quantity scales as a result of data 'frequency' differences between model training sets). This difference also forms a lower bound on the total-variation divergence $TV(p, q)$ between models $p, q$, where $TV(p, q) \geq |r_p - r_q| \implies 1 - \alpha_T \geq |r_p - r_q|$ (by the reverse triangle-inequality). This directly implies that when data representation differences are high (i.e., $|r_q-r_p|$ is high), the divergence $TV(p, q)$ is constrained to be high, worsening speed-up.
>
> **Q6** *"I understand that the proposed mitigation strategy will trade-off speed-up gains across tasks. ... After using the proposed procedure, what happens to the speed-up on the English task? Looking at the proposed gradient in Sec 7, L258, weighting across  tasks is uniform . For a dominant task whose base occurrence probability is above , it will be weighted down. Is this correct?"*
>
> This is definitely an important point that is worth expanding on. The first note to establish is that speed-up unfairness mitigation is not 'zero-sum'. If the alignment procedure is designed correctly, and model capacity is high, it is possible to raise speed-up across all tasks simultaneously (see Tables 1 and 2). Our goal is thus to reduce unfairness without forcibly degrading speed-up on any task. Our proposed algorithm takes a step towards this goal. Namely, the balancing effect of our proposed gradient $(D_T - D_{\min})\nabla_\theta D_T$ comes from the scaling term $w_T \triangleq (D_T - D_{\min})$. The term $w_T$ acts as a weight for task $T$, that scales the distillation correction ($\nabla_\theta D_T$) by the magnitude of unfairness observed for $T$; ($D_T - D_{\min}$). This means that for a majority task, say English, the weight $w_{\text{en}}=(D_T - D_{\min})=(D_{\text{en}} - D_{\text{en}})=0$, results in no distillation update to English.
>
> However, applying our gradient stochastically (as for s-CDF), means that the majority task $T_{\min}$ is recomputed each mini-batch (i.e., the fastest task within the mini-batch). This is both (i) more practical for determining $T_{\min}$ (as no large scale evaluation is required prior to optimization) and (ii) produces a natural self-correction if $T_{\min}$ (i.e., English) starts degrading during. To elaborate on point (ii), observe in Figure 9, where the clear majority task 'English', is determined to be $T_{\min}$ in only $\sim93\%$ of mini-batches, meaning in $\sim7\%$ of mini-batches, English still receives some update ($w_{\text{en}}\nabla_\theta D_{\text{en}}$). If English were to degrade too much during optimization, it would be less likely to be sampled as the fastest task $T_{\min}$, increasing the expected update. This is further supported by our ablations in (Tables 1, 2), where the majority task still increases in speed-up during s-CDF.
>
> Comparatively, the degrading gradient $-(D_T - D_{\min})\nabla_\theta D_T$ reduces unfairness via directly worsening the speed of $T_{\min}$, to approach the slower tasks. We see that this instead causes degradation in the fastest task, Tables 1, and 2, which is contrary to our mitigation goals. Additionally, uniform weighting ($w_T=1; \forall T$) is not optimal for targeting unfairness, and allocates too much weight to the faster tasks (Tables 1, and 2). We would lastly like to re-emphasize that our work is not centered around comparing differing correction mechanisms, and leaves this open for the community to build on. We have rather tried to illustrate the substantive nature of the unfairness problem, and provide examples of promising, and principled mitigation strategies compatible with our novel framework.
>
> **Q7** *"Is it correct that the proposed fairness mitigation strategy does not change the overall quality (across all tasks)?..."*
>
> Yes this is absolutely accurate. We agree that this may be worth emphasizing for unfamiliar reader. We have incorporated results in Appendix C (see Figure 11) to showcase that verifier-loss $r_p$ is frozen during s-CDF, which paired with our proof in Appendix A.5 theoretically guarantees no quality degradation.
>
> ---
>
> Thank you for your thoughtful review! We hope that our responses have fully addressed your concerns and merit your strong support for this work, and we remain happy to provide any additional clarification if needed.
>
>
> **References**
> [Zhou et al. 2024]: Yongchao Zhou et al. "DistillSpec: Improving Speculative Decoding via Knowledge Distillation"; ICLR 2024

---

> ### Author Response · Authors · 2025-11-25
>
> Dear Reviewer,
>
> as the discussion period is nearing its end, we wanted to ask if there are any follow-up points we can clarify.
>
> We believe we have responded to all questions and concerns raised, in addition to taking a few days to run the additional experiments necessary to demonstrate our points, all of which are incorporated in the revised version of the paper.
> \
> If there are no further points of clarification regarding the manuscript, we kindly ask that reviewer XiyQ consider increasing their score to reflect the improvements and clarifications we have provided. We are happy to continue to engage in discussion and answer any questions!

---

### Meta-Review · Area_Chair_inQc · 2026-01-07

**Summary:**

The initial reviews for this submission were generally bellow the borderline. Common concerns included:
1. Empirical Robustness: Reviewers (XiyQ, y91N) felt the initial experiments were too limited, focusing primarily on a bi-lingual (English vs. Japanese) setup or a specific model series.
2. Missing Baselines: A lack of comparison against other potential mitigation strategies, such as DistillSpec or standard data balancing, was noted (XiyQ, 4XQp).
3. Generalizability: Questions were raised about how these findings scale to larger verifier models or generalize beyond natural language tasks.
Overall this submission is a borderline case

**Reviewer Concerns:**

Addressed Concerns:
1. Expanded Empirical Evidence: The authors broadened their experimental scope, including evaluations across seven languages (up from two) and scaling the verifier model up to 32B parameters.
2. Ablative Baselines: The authors added comparisons against DistillSpec and several architectural variants of their s-CDF method.

Outstanding Concerns:
1. Broader Task Definitions: While the authors provided a theoretical proof for overlapping tasks, the empirical focus remains on natural language subsets. Exploration of other "task" definitions (e.g., code vs. prose) remains suggested for future work.
2. The Novelty and concerns about the contribution of the submission still exist after rebuttal.

**Reviewer Scores:**

XiyQ	4->4/6  The new baselines and scaling experiments directly addressed the "Soundness" concerns.
4XQp	4->4.   The concerns about contributions still exist.
y91N	4->6	Reviewer explicitly stated satisfaction with the 32B scaling and language expansion.
zdoT	6 ->6	Positive. Rebuttal reinforced the practical social impact and user-experience arguments.

---

### Decision · Program_Chairs · 2026-01-26

Reject